# Variational Inference with Mixtures of Isotropic Gaussians

**Marguerite Petit-Talamon**
CREST, ENSAE
Institut Polytechnique de Paris
`marguerite.petittalamon@ensae.fr`

**Marc Lambert**
INRIA, Ecole Normale Supérieure
DGA, French Procurement Agency
`marc.lambert@inria.fr`

**Anna Korba**
CREST, ENSAE
Institut Polytechnique de Paris
`anna.korba@ensae.fr`

## Abstract

Variational inference (VI) is a popular approach in Bayesian inference, that looks for the best approximation of the posterior distribution within a parametric family, minimizing a loss that is typically the (reverse) Kullback-Leibler (KL) divergence. In this paper, we focus on the following parametric family: mixtures of isotropic Gaussians (i.e., with diagonal covariance matrices proportional to the identity) and uniform weights. We develop a variational framework and provide efficient algorithms suited for this family. In contrast with mixtures of Gaussian with generic covariance matrices, this choice presents a balance between accurate approximations of multimodal Bayesian posteriors, while being memory and computationally efficient. Our algorithms implement gradient descent on the location of the mixture components (the modes of the Gaussians), and either (an entropic) Mirror or Bures descent on their variance parameters. We illustrate the performance of our algorithms on numerical experiments.

## 1 Introduction

The core problem of Bayesian inference is to sample from a posterior distribution $\pi$ over model parameters, combining prior knowledge with observed data. Unfortunately, the posterior distribution is generally difficult to compute due to the presence of an intractable integral (the normalization constant), and its density with respect to the Lebesgue measure on $\mathbb{R}^d$, also denoted $\pi$, is only known in unnormalized form as $\pi \propto e^{-V}$. Variational inference (VI, [Blei et al., 2017]) is a prominent alternative to standard Markov chain Monte Carlo (MCMC, [Roberts and Rosenthal, 2004]) that approximates the posterior by optimizing over a family of tractable distributions. While this restriction can introduce bias, VI is typically much faster than MCMC, since it reframes the sampling problem as a (generally finite-dimensional) optimization one over the parameters of the variational family. Specifically, VI seeks a distribution in a parametric family $\mathcal{C}$ that minimizes a discrepancy to the target posterior, typically the reverse Kullback-Leibler (KL) divergence:

$$\min_{\mu \in \mathcal{C}} \mathrm{KL}(\mu|\pi), \tag{1}$$

where $\mathrm{KL}(\mu|\pi) = \int \log(d\mu/d\pi)d\mu$ if $\mu$ is absolutely continuous with respect to $\pi$ denoting $d\mu/d\pi$ its Radon-Nikodym density, and $+\infty$ else.

There exist various choices of variational families and suited algorithms in the literature of VI. For instance, Mean-field variational inference (MFVI) aims to find an approximate posterior that

factors as a product of distributions [Lacker, 2023]. Recent studies by Arnese and Lacker [2024] and Lavenant and Zanella [2024] have investigated solving this problem through the lens of coordinate ascent variational inference (CAVI, Bishop [2006]), or over polyhedral sets with first-order algorithms [Jiang et al., 2024]. Traditionally, the variational family is parameterized by a finite-dimensional set of parameters, yet, Yao and Yang [2022], Yao et al. [2024] have extended this framework to MFVI over infinite-dimensional families. Alternatively, Gaussian variational inference [Barber and Bishop, 1997, Opper and Archambeau, 2009] has garnered significant attention due to its computational tractability and theoretical appeal [Challis and Barber, 2013]. More recently, a new class of Gaussian VI algorithms has emerged, based on the discretization of the gradient flow of the KL divergence on the space of Gaussian measures equipped with the Wasserstein-2 metric: Lambert et al. [2022] proposed a forward (explicit) time discretization approach, while Diao et al. [2023], Domke et al. [2023] introduced a forward-backward splitting scheme inspired by [Salim et al., 2020]. Other algorithms were recently introduced for Gaussian VI by optimizing a weighted score-based (Fisher) divergence [Modi et al., 2025, Cai et al., 2024b]. These methods bring valuable insights for Gaussian VI, but may provide a too crude approximation of the target distribution if the latter is multimodal. In this setting, deep generative approaches such as normalizing flows [Tabak and Vanden-Eijnden, 2010, Papamakarios et al., 2021, Kobyzev et al., 2020] offering tractable, normalized densities and efficient sampling, have emerged as flexible and powerful alternatives to traditional Gaussian or factorized variational families. While it offers a flexible and widely applicable approach to VI, it is not tailored to a specific variational family. Moreover, despite their ability to represent multimodal distributions using expressive variational families with neural networks, it can suffer from mode collapse [Soletskyi et al., 2024], especially as the dimension increases, as shown recently in large scale empirical evaluation from [Blessing et al., 2024].

A particularly compelling variational family is the one of mixtures of Gaussians. Indeed, the latter can capture complex, multimodal distributions. Also, they lead to tractable approximate posteriors, since sampling mixtures is computationnally cheap, and also marginalizations, or expectations of linear and quadratic functions can be computed in closed-form. For this variational family, several algorithms have been proposed [Lin et al., 2019, Arenz et al., 2018, 2023] based on natural gradient descent over the natural parameters of the Gaussians. An extension of the Wasserstein gradient flow approach for Gaussian mixtures has also been proposed in Lambert et al. [2022]. However, their practical utility is often hindered by the computational challenges of inference, particularly when handling high-dimensional distributions. Indeed, a key challenge in deploying Gaussian mixture models lies in parameterizing the covariance matrices. While full covariance matrices provide maximum flexibility, their quadratic scaling with dimensionality leads to high computational demands, even if some solutions have been proposed to store them efficiently [Challis and Barber, 2013, Bonnabel et al., 2024]. To address this issue, we restrict the variational family of mixtures of Gaussians to diagonal, more precisely isotropic covariance matrices (i.e., assuming equal variance across all dimensions), with uniform weights. Mode collapse may result from mean alignment, i.e., two components' means may align with the same mode of the target, instead of covering multiple modes; and vanishing weights [Soletskyi et al., 2024]. Using fixed weights, as we do, prevents the latter. With this structure, each Gaussian component in the mixture is parameterized by $d + 1$ parameters in dimension $d$. As a result, a mixture of $N$ isotropic Gaussians requires a memory cost of $N(d + 1)$, compared to $N(d^2 + d)$ for a full covariance mixture. Then, optimizing mixtures of isotropic Gaussians incurs a memory cost roughly equivalent to that of optimizing the means only. We show in this study that this choice of variational family balances between accuracy of the variational approximation, i.e., its ability to model multimodal target distributions, and the computational efficiency of the associated algorithms.

Our contributions include the development of a variational framework and algorithms tailored to isotropic Gaussian mixtures, as well as an empirical evaluation across synthetic and real-world datasets, demonstrating that our approach achieves a compelling balance between modeling accuracy for multimodal targets and computational efficiency. This paper is organized as follows. Section 2 provides the relevant background on the geometry of the space of isotropic Gaussians, and on optimization schemes based on the Bures and entropic mirror descent geometries. In Section 3, we introduce the general setting for optimization over mixtures of isotropic Gaussians with uniform weights. Section 4 presents our algorithms to efficiently optimize over this variational family. In Section 5 we discuss related work in the Variational Inference literature. Our numerical results are to be found in Section 6.

**Notation.** We write a Gaussian distribution on $\mathbb{R}^d$ with mean $m$ and variance $\epsilon$ as $\mathcal{N}(m, \epsilon I_d)$, where $I_d$ denotes the d-dimensional identity matrix; and $\mathcal{N}(x; m, \epsilon I_d)$ its density evaluated at $x$. We denote by $\mathcal{P}_2(\mathbb{R}^d)$ the set of probability distributions on $\mathbb{R}^d$ with bounded second moments. Consider $\mu, \nu \in \mathcal{P}_2(\mathbb{R}^d)$, the Wasserstein-2 ($W_2$) distance is defined as $W_2^2(\mu, \nu) = \inf_{s \in \mathcal{S}(\mu, \nu)} \int \|x - y\|^2 ds(x, y)$, where $\mathcal{S}(\mu, \nu)$ is the set of couplings between $\mu$ and $\nu$. We will denote $k_\epsilon$ the normalized Gaussian kernel on $\mathbb{R}^d$ with variance $\epsilon$, i.e. $k_\epsilon(x) = (2\pi\epsilon)^{-d/2} \exp(-\|x\|^2/(2\epsilon))$. For $\mu \in \mathcal{P}_2(\mathbb{R}^d)$, we denote by $k_\epsilon \star \mu$ its convolution with the Gaussian kernel, that writes $k_\epsilon \star \mu = \int k_\epsilon(\cdot - x) d\mu(x)$. We denote $\mathrm{Tr}$ the trace function.

## 2 Preliminaries (Gaussian VI)

This section introduces key concepts on the space of isotropic Gaussians, as well as different (time-discretized) gradient flows one can consider through the Bures-Wasserstein or entropic mirror descent geometries.

### 2.1 Isotropic Gaussians (IG)

The space of isotropic Gaussians is defined as $\mathrm{IG} = \{\mathcal{N}(m, \epsilon I_d), m \in \mathbb{R}^d, \epsilon \in \mathbb{R}^{+*}\}$ and is a subspace of $\mathcal{P}_2(\mathbb{R}^d)$. When equipped with the $W_2$ distance, this space has a particularly tractable geometric structure. Indeed, the $W_2$ distance between two isotropic Gaussians $\mathcal{N}(m, \epsilon I_d), \mathcal{N}(m', \tau I_d)$ takes the form of a Bures-Wasserstein (BW) distance:

$$\mathrm{BW}^2(\mathcal{N}(m, \epsilon I_d), \mathcal{N}(m', \tau I_d)) = \|m - m'\|^2 + \mathrm{B}^2(\epsilon \mathrm{Id}, \tau \mathrm{Id}), \tag{2}$$

where $\mathrm{B}$ denotes the Bures metric [Bhatia et al., 2019] between positive definite matrices and $\mathrm{B}^2(\epsilon \mathrm{Id}, \tau \mathrm{Id}) = d(\epsilon + \tau - 2\sqrt{\epsilon\tau})$. This formula reflects the separable nature (with respect to the means and variances) of the Wasserstein metric on the space of isotropic Gaussians IG. Interestingly, the metric space $(\mathrm{IG}, \mathrm{BW})$ of isotropic Gaussians equipped with the BW distance can be seen as a submanifold of the space of (all) Gaussian distributions equipped with the same metric, which can itself be seen as a submanifold of the Wasserstein space $(\mathcal{P}_2(\mathbb{R}^d), W_2)$. Indeed, the BW geodesic between $\mu, \nu \in \mathrm{IG}$ also lies in IG, see Appendix A.1.

### 2.2 Bures-Wasserstein gradient descent on IG

Recall that our goal is to minimize a functional objective $\mathrm{KL}(\cdot|\pi)$ as defined in Eq (1), where $\pi \propto e^{-V}$, firstly on IG in this section, before being able to tackle mixtures of isotropic Gaussians. In this subsection we explain how to derive a gradient flow with respect to the Bures-Wasserstein geometry, and provide a discrete optimization scheme. To this goal, we first define a minimizing movement scheme on IG. For $p_0 \in \mathrm{IG}$ and $\gamma > 0$ a step-size, define:

$$p_{k+1} = \underset{p \in \mathrm{IG}}{\arg\min} \left\{ \mathrm{KL}(p|\pi) + \frac{1}{2\gamma} \mathrm{BW}^2(p, p_k) \right\}, \tag{3}$$

which corresponds to a JKO scheme Jordan et al. [1998], but where the solution is constrained to lie in IG. In the limit $\gamma \to 0$, we obtain a Wasserstein gradient flow of measures projected on IG, i.e. a continuous curve $(p_t)_t \in \mathrm{IG}$ decreasing the KL, and which is governed by differential equations for the mean $(m_t)_t$ and variance $(\epsilon_t)_t$ (see Appendix A.3). Such a flow can exhibit a favorable dynamical behavior under a strong log-concavity assumption on the target distribution, as demonstrated in the following proposition.

**Proposition 2.1.** *Suppose that $\nabla^2 V \succeq \alpha I_d$ for some $\alpha \in \mathbb{R}$. Then, for any $p_0 \in \mathrm{IG}$, there is a unique solution $(p_t)_t$ to the flow obtained as a limit of Eq (3) as $\gamma \to 0$. Then, for all $t \geq 0$ and $p^* \in \mathrm{IG}$,*

$$\mathrm{KL}(p_t|\pi) - \mathrm{KL}(p^*|\pi) \leq e^{-2\alpha t} \{ \mathrm{KL}(p_0|\pi) - \mathrm{KL}(p^*|\pi) \},$$

*implying that the flow converges linearly when $\alpha > 0$.*

The full proof of Proposition 2.1 is provided in Appendix A.2, and is a direct application of the one of [Lambert et al., 2022, Corollary 3]. It relies on the fact that $\mathrm{KL}(\cdot|\pi)$ is an $\alpha$-convex objective functional along $W_2$ geodesics when the target potential $V$ is $\alpha$-convex, see e.g. [Villani, 2009,

Theorem 17.15]; and since $W_2$ (equivalently BW) geodesics between isotropic Gaussians lie in IG (see Appendix A.1), the objective is also convex on (IG, BW). This result indicates that strongly log-concave targets can be efficiently approximated using isotropic Gaussians.

Here, we propose to evaluate an explicit time-discretization of this gradient flow, as it is computationally less expensive than an implicit scheme such as Eq (3). To this end, let $F : \mathbb{R}^d \times \mathbb{R}^{+*}$ defined as $F(m, \epsilon) := \mathrm{KL}(\mathcal{N}(m, \epsilon \mathrm{I}_d)|\pi)$. Starting from some $\theta_0 = (m_0, \epsilon_0) \in \mathbb{R}^d \times \mathbb{R}^{+*}$, we consider the following scheme:

$$\theta_{k+1} = \underset{\theta \in \mathbb{R}^d \times \mathbb{R}^{+*}}{\operatorname{argmin}} \left\{ \langle \nabla F(\theta_k), \theta - \theta_k \rangle + \frac{1}{2\gamma} \mathrm{BW}^2(\mathcal{N}_\theta, \mathcal{N}_{\theta_k}) \right\}, \quad (4)$$

denoting $\theta := (m, \epsilon)$ and $\mathcal{N}_\theta := \mathcal{N}(m, \epsilon \mathrm{I}_d)$. Note that the scheme above is similar to Eq (3) but where the objective has been linearized. Thanks to the decomposition of the BW distance given in Eq (2), it leads to the following updates on the mean and variances:

$$m_{k+1} = m_k - \gamma \nabla_m F(m_k, \epsilon_k)$$
$$\epsilon_{k+1} = \left( 1 - \frac{2\gamma}{d} \nabla_\epsilon F(m_k, \epsilon_k) \right)^2 \epsilon_k, \quad (5)$$

where $\nabla_m F(m_k, \epsilon_k) = \mathbb{E}_{p_k}[\nabla V]$ and $\nabla_\epsilon F(m_k, \epsilon_k) = \frac{1}{2} \left( \frac{1}{d\epsilon_k} \mathbb{E}_{p_k}[(\cdot - m_k)^\top \nabla V] - \frac{1}{\epsilon_k} \right)$. We observe that while the first update on the mean is a simple gradient descent, the latter update ensures that the variance remains positive and differs from a simple Euler discretization of the associated differential equation (see Appendix A.3). We provide the details of the computation as well as an interpretation of these updates as a Riemannian gradient descent in Appendix A.4.

## 2.3 Entropic Mirror Descent on IG

We now turn to an alternative descent scheme on IG, namely mirror descent, which relies on a geometry different from the $W_2$ one described in the previous subsection. Mirror descent is an optimization algorithm that was introduced to solve constrained convex problems [Nemirovskij and Yudin, 1983], and that uses in the optimization updates a cost (or "geometry") that is a Bregman divergence [Bregman, 1967], whose definition is given below.

**Definition 2.2.** Let $\phi : \mathcal{X} \to \mathbb{R}$ a strictly convex and differentiable functional on a convex set $\mathcal{X}$, referred to as a Bregman potential. The $\phi$-Bregman divergence is defined for any $x, y \in \mathcal{X}$ by:

$$B_\phi(y|x) = \phi(y) - \phi(x) - \langle \nabla \phi(x), y - x \rangle.$$

Further details on mirror descent and its connection to standard algorithms such as gradient descent are provided in Appendix A.5.

Hence, we propose to choose an appropriate Bregman divergence on the space of covariance matrices, namely a generalized Kullback-Leibler divergence between positive definite matrices: $\overline{\mathrm{KL}}(A|B) = \mathrm{Tr}(A(\log A - \log B)) - \mathrm{Tr}(A) + \mathrm{Tr}(B)$. The latter object, also called Von Neumann relative entropy, is a Bregman divergence whose Bregman potential is the Von Neumann entropy $\phi : A \mapsto \mathrm{Tr}(A \log A)$ (and where $\langle A, B \rangle = \mathrm{Tr}(AB)$). Note that $\overline{\mathrm{KL}}(\epsilon \mathrm{I}_d | \tau \mathrm{I}_d) = d \left( \epsilon \log \frac{\epsilon}{\tau} - \epsilon + \tau \right)$. Then, we can define a descent scheme on IG as follows, starting from some $\theta_0 = (m_0, \epsilon_0) \in \mathbb{R}^d \times \mathbb{R}^{+*}$:

$$\theta_{k+1} = \underset{\theta \in \mathbb{R}^d \times \mathbb{R}^{+*}}{\operatorname{argmin}} \left\{ \langle \nabla F(\theta_k), \theta - \theta_k \rangle + \frac{1}{2\gamma} \|m - m_k\|^2 + \frac{1}{2\gamma} \overline{\mathrm{KL}}(\epsilon \mathrm{I}_d | \epsilon_k \mathrm{I}_d) \right\},$$

denoting again $\theta = (m, \epsilon)$. Note that compared to the scheme of Eq (4), only the update on the variance differs, and is given by:

$$\epsilon_{k+1} = \epsilon_k \exp\left( -\frac{2\gamma}{d} \nabla_\epsilon F(m_k, \epsilon_k) \right), \quad (6)$$

see Appendix A.6 for the computations. This update, as the one in Eq (5), also guarantees that the variance parameter $\epsilon$ remains strictly positive; and is known as entropic mirror descent [Beck and Teboulle, 2003][1].

---

[1][Beck and Teboulle, 2003] used this exponential update followed by a renormalization to optimize over the simplex.

## 3 Mixtures of Isotropic Gaussians (MIG)

We now turn to the problem of optimizing the KL objective to the target distribution $\pi$ over the family of mixtures of isotropic Gaussians. We will consider the VI problem Eq (1) for a specific setting where the variational family is the set of mixtures of $N$ isotropic Gaussians, for some $N \in \mathbb{N}^*$, with equally weighted components:

$$\mathcal{C}^N = \Big\{ \frac{1}{N} \sum_{j=1}^N \mathcal{N}(m^j, \epsilon^j \mathrm{I}_d), \ [m^j, \epsilon^j]_{j=1}^N \in (\mathbb{R}^d \times \mathbb{R}^{+*})^N \Big\}.$$

Note that any distribution $\nu \in \mathcal{C}^N$ writes $\nu = \frac{1}{N} \sum_{j=1}^N k_{\epsilon^j} \star \delta_{m^j}$, for some $[m^j, \epsilon^j]_{j=1}^N \in (\mathbb{R}^d \times \mathbb{R}^{+*})^N$, where $k_{\epsilon^j}$ is the Gaussian kernel with variance $\epsilon^j$ and $\delta_{m^j}$ is the Dirac at $m^j$. Then, we define our loss function $F : (\mathbb{R}^d)^N \times (\mathbb{R}^+)^N \to \mathbb{R}^{+*}$ as:

$$F([m^j, \epsilon^j]_{j=1}^N) := \mathrm{KL}\Big( \frac{1}{N} \sum_{j=1}^N \mathcal{N}(m^j, \epsilon^j \mathrm{I}_d) \Big| \pi \Big). \tag{7}$$

The following proposition provides useful formulas regarding the gradients of this objective.

**Proposition 3.1.** *Let $\mu = \frac{1}{N} \sum_{j=1}^N \delta_{m^j}$ and denote $k_\epsilon \otimes \mu = \frac{1}{N} \sum_{j=1}^N k_{\epsilon^j} \star \delta_{m^j}$. Assume $\pi \in \mathcal{C}^1(\mathbb{R}^d)$. The gradients of $F$ with respect to $m^j, \epsilon^j \in \mathbb{R}^d \times \mathbb{R}^{+*}$ write:*

$$\nabla_{\epsilon^j} F([m^j, \epsilon^j]_{j=1}^N) = \frac{1}{2N\epsilon^j} \mathbb{E}_{k_{\epsilon^j} \star \delta_{m^j}} \Big[ (\cdot - m^j)^T \nabla \ln\Big( \frac{k_\epsilon \otimes \mu}{\pi} \Big)(\cdot) \Big],$$

$$\nabla_{m^j} F([m^j, \epsilon^j]_{j=1}^N) = \frac{1}{N} \mathbb{E}_{k_{\epsilon^j} \star \delta_{m^j}} \Big[ \nabla \ln\Big( \frac{k_\epsilon \otimes \mu}{\pi} \Big)(\cdot) \Big].$$

The proof of Proposition 3.1 can be found in Appendix A.7. Note that the means and variances in the mixture interact through the terms $\nabla \ln(k_\epsilon \otimes \mu)$ in the gradients. Remarkably, our computations provide an expression of the gradient with respect to the variance that only involves a scalar product $(\cdot - m^j)^T \nabla \ln \Big( \frac{k_\epsilon \otimes \mu}{\pi} \Big)$, which can be computed efficiently with a computational cost in $\mathcal{O}(d)$. In practice, the expectations over the Gaussian components $k_{\epsilon^j} \star \delta_{m^j}$ for $j = 1, \ldots, N$ will be estimated with Monte Carlo integration.

## 4 Algorithms for VI on MIG

### 4.1 General optimization framework

We propose, for the optimization of the objective $F$ on $(\mathbb{R}^d)^N \times (\mathbb{R}^{+*})^N$ defined in Eq. (7), or equivalently $\mathrm{KL}(\cdot|\pi)$ on $\mathcal{C}^N$, to perform joint optimization on the means and variances of the mixture. This joint optimization involves a gradient descent update on the means, and either a Bures or entropic mirror descent update on the variances. Our approach is summarized in Algorithm 1.

---

**Algorithm 1** MIG optimization with IBW or MD

---

  **Input:** initial means and variances $(m_0^j, \epsilon_0^j)_{j=1}^N$, step-size $\gamma$, number of iterations $T$.
  **for** $k = 1$ **to** $T$ **do**
    **for** $i = 1$ **to** $N$ **do**
      Update $m_{k+1}^i = m_k^i - \gamma N \nabla_{m_k^i} F\left([m_k^j, \epsilon_k^j]_{j=1}^N\right)$ (GD)
      Update $\epsilon_k^i$ with IBW (Eq. 12) or MD (Eq. 13)
    **end for**
  **end for**

---

We now describe the optimization of the variance parameters using either Bures or entropic mirror descent updates in the next subsections. These methods rely on careful adaptations of the schemes introduced in Section 2, originally defined for a single isotropic Gaussian, to the mixture setting.

## 4.2 Bures (IBW) update

This section extends the framework of Section 2.2 to Gaussian mixtures and presents a new formulation of the JKO scheme adapted to this setting. The $W_2$ distance between two Gaussian mixtures is intractable and does not admit a closed form, in contrast with the BW distance on IG. Then, we cannot obtain direct updates on the mean and variances for a JKO scheme restricted to $\mathcal{C}^N$. To address this issue, we will represent a mixture with its associated mixing measure. Lambert et al. [2022] proposed a similar approach to derive a Wasserstein gradient flow on Gaussian mixtures. However, they considered a mixing measure with an infinite number of components and did not provide a formal derivation of a fully explicit discrete (in time and space) scheme.

Any uniform-weight Gaussian mixture $\nu \in \mathcal{C}^N$ can be identified to a mixing measure $\hat{p} = \frac{1}{N}\sum_{j=1}^{N}\delta_{(m^j,\epsilon^j)}$ on $(\mathbb{R}^d \times \mathbb{R}^{+*})^N$ where for any $x \in \mathbb{R}^d$ we have:

$$\nu(x) = \int \mathcal{N}(x; m, \epsilon \mathrm{I}_d) d\hat{p}(m, \epsilon) \tag{8}$$

Note that the space $\mathcal{C}^N$ allows for the full identification of a mixture, up to a reordering of the indices, since the corresponding mixing measure contains only $N$ particles with equal weights. This avoids the identifiability issues that arise when dealing with a mixing measure supported on a continuous (infinite) set of components, which constitutes an overparameterized model, see [Chewi et al., 2024, Section 5.6]. See also Appendix B.1 for more details. Following Chen et al. [2019], we first consider a Wasserstein distance between mixing measures denoted $W_{bw}$, where the cost is a squared Bures-Wasserstein distance, i.e. $c((m, \epsilon), (m', \tau)) = \mathrm{BW}^2(\mathcal{N}(m, \epsilon \mathrm{Id}), \mathcal{N}(m', \tau \mathrm{Id}))$. We then construct the JKO scheme on mixing measures at each step $k$ as:

$$\hat{p}_{k+1} = \underset{(m^j, \epsilon^j)_{j=1}^N}{\arg\min} \left\{ \mathrm{KL}(\nu|\pi) + \frac{1}{2\gamma} W_{bw}^2(\hat{p}, \hat{p}_k) \right\}, \tag{9}$$

where $\hat{p} = \frac{1}{N}\sum_{j=1}^{N}\delta_{(m^j,\epsilon^j)}$, $\hat{p}_k = \frac{1}{N}\sum_{j=1}^{N}\delta_{(m_k^j,\epsilon_k^j)}$ and $\nu$ is defined as in Eq (8). The resulting Gaussian mixture is $\nu_{k+1} = \int \mathcal{N}(m, \epsilon \mathrm{I}_d) d\hat{p}_{k+1}(m, \epsilon)$. Since $\hat{p}$ and $\hat{p}_k$ are two discrete measures with an equal number of components, the above Wasserstein distance simplifies as:

$$W_{bw}^2(\hat{p}, \hat{p}_k) = \min_{\sigma} \frac{1}{N}\sum_{j=1}^{N} \mathrm{BW}^2(\mathcal{N}(m^j, \epsilon^j \mathrm{I}_d), \mathcal{N}(m_k^{\sigma(j)}, \epsilon_k^{\sigma(j)} \mathrm{I}_d)), \tag{10}$$

where $\sigma$ is a permutation of the $N$ indices in the mixture. Solving the JKO scheme Eq (9) is now tractable and we can compute the limiting flow as $\gamma \to 0$, since at the limit $\sigma(i) = i$. The continuous-time equations of the flow in the isotropic case are given in Appendix B.2. They match the continuous-time equations for the means and covariances derived in [Lambert et al., 2022, Section 5.2] and recalled in Appendix E.1.

Similarly to Section 2.2, we consider an explicit time-discretization of this flow, using a linearization of the objective in Eq (9). This leads us to the scheme:

$$[\theta_{k+1}]_{j=1}^N = \underset{(\theta^j)_{j=1}^N}{\arg\min} \left\{ \langle \nabla F([\theta_k^j]_{j=1}^N), [\theta^j]_{j=1}^N - [\theta_k^j]_{j=1}^N \rangle + \frac{1}{2\gamma N}\sum_{j=1}^{N} \mathrm{BW}^2(\mathcal{N}_{\theta^j}, \mathcal{N}_{\theta_k^j}) \right\}, \tag{11}$$

assuming that $\sigma(i) = i$ in Eq (10) for $\gamma$ small enough. Finally, the variance updates for the Gaussian components are:

$$\boxed{\textbf{IBW update} \quad \text{For } j = 1, \ldots, N: \quad \epsilon_{k+1}^j = \left(1 - \frac{2N\gamma}{d}\nabla_{\epsilon^j} F([m_k^j, \epsilon_k^j]_{j=1}^N)\right)^2 \epsilon_k^j.} \tag{12}$$

The update on the means takes the form of Eq (GD). Details of the computations are deferred to Appendix B.3. Ultimately, we obtain a system of Gaussian particles $(m^j, \epsilon^j)_{j=1}^N$ that interact through the gradient of the objective.

*Remark* 4.1. Note that we can characterize the discrepancy between our JKO scheme on the mixing measure and the original JKO scheme restricted to Gaussian mixtures. Indeed, the Wasserstein distance on the mixing measure is related to the $W_2$ distance on $\mathcal{P}_2(\mathbb{R}^d)$ as follows: $0 \leq W_{bw}^2(\hat{p}, \hat{p}_k) - W_2^2(\nu, \nu_k) \leq 2\sqrt{2d\epsilon^*}$ where $\epsilon^*$ is the maximal variance of the mixtures $\nu, \nu_k$. This result is a direct consequence of [Delon and Desolneux, 2020, Proposition 6], see Appendix B.4 for further details.

## 4.3 Entropic mirror descent (MD) update

In this section we provide an alternative way to optimize the variances of the mixture, based on mirror descent ideas introduced in Section 2.3. In particular, generalizing to $N$ components what we have done for Eq (6), and by analogy with the scheme Eq (11), we consider the following scheme:

$$[\theta_{k+1}^j]_{j=1}^N = \underset{(\theta^j)_{j=1}^N}{\arg\min}\Big\{\langle \nabla F([\theta_k^j]_{j=1}^N), [\theta^j]_{j=1}^N - [\theta_k^j]_{j=1}^N\rangle + \frac{1}{2\gamma N}\sum_{j=1}^N \|m^j - m_k^j\|^2 + \overline{\mathrm{KL}}(\epsilon^j \mathrm{I}_d|\epsilon_k^j \mathrm{I}_d)\Big\}.$$

Then, at step $k \geq 0$, the udpate on the variances takes the form:

$$\boxed{\textbf{MD update}\quad \text{For } j = 1, \ldots, N: \quad \epsilon_{k+1}^j = \epsilon_k^j \exp\Big(-\frac{2N\gamma}{d}\nabla_{\epsilon^j}F([m_k^j, \epsilon_k^j]_{j=1}^N)\Big),} \tag{13}$$

while the update on the means remains Eq (GD), see Appendix A.6 for the detailed computations.

## 5 Related Work

In this section, we provide an overview of relevant work on VI with mixtures of Gaussians.

Several studies have addressed VI for mixture models, emphasizing computational aspects. Gershman et al. [2012] optimize an approximate ELBO using L-BFGS (a quasi-Newton method), relying on successive approximations of ELBO terms for mixtures of Gaussians. However, while the original KL objective in VI defines a valid divergence between probability distributions, their optimization objective departs significantly from it.

Lin et al. [2019], Arenz et al. [2018] propose natural gradient descent (NGD) updates on the natural parameters of the Gaussians for each component of the mixture, and on the categorical distribution over weights. These methods are unified and extended in [Arenz et al., 2023], which introduces computational improvements. Natural gradient descent differs from standard gradient descent by performing steepest descent with respect to changes in the underlying distribution, measured using the Fisher information metric. In other words, the natural gradient is the standard gradient preconditioned by the inverse of the Fisher information matrix. For exponential families, such as Gaussians, the natural gradient of the objective with respect to natural parameters coincides with the standard gradient of the (reparametrized) objective when expressed in terms of expectation parameters (i.e., the moments of the Gaussians). This has some pleasant consequences, including closed-form updates on means and covariances, since the natural parameter admits a simple expression in terms of means and variances for Gaussians. The NGD updates (fixing the weights of the mixture) on means and variances write:

$$\frac{1}{\epsilon_{k+1}^i} - \frac{1}{\epsilon_k^i} = \frac{2N\gamma}{d}\nabla_{\epsilon_k^i}F([m_k^j, \epsilon_k^j]_{j=1}^N), \quad m_{k+1}^i - m_k^i = -\epsilon_{k+1}^i N\gamma \nabla_{m_k^i}F([m_k^j, \epsilon_k^j]_{j=1}^N). \tag{14}$$

We refer to Appendix C for more details and the computations. *In particular, the latter update does not guarantee that the variances remain positive, and the update on the mean is multiplied by the current covariance.* In contrast, our updates on the means and covariances are decoupled (except through the gradient of the objective), and our updates on variances enforce positivity.

[Huix et al., 2024] considered the setting where the variances of the mixture are shared, equal to $\epsilon \mathrm{I}_d$ with $\epsilon \in \mathbb{R}^{+*}$ that is kept fixed, and only the means $(m^1, \ldots, m^N)$ are optimized with gradient descent (GD). In that setting, they proved a descent lemma showing that the KL objective functional decreases along the GD iterations, under some conditions including a maximal step-size, a boundedness conditions on the second moment of the (distribution) iterations and a finite number of components $N$. They also provided an upper bound on the approximation error, i.e. the minimal KL divergence between a $N$-component mixture of Gaussians with uniform weights and shared isotropic covariance matrices between components is upper bounded as $\mathcal{O}(\frac{\log(N)}{N})$, when $\pi$ writes as an infinite mixture of these components. Interestingly, this bound is valid for mixtures of isotropic Gaussians with different variances, as we show in Appendix D. Yet, fixing the covariances to a constant factor $\epsilon$ of the identity, as in [Huix et al., 2024], limits a lot the expressiveness of the variational family. Hence, we focus in this work on the more general variational family defined by $\mathcal{C}^N$.

# 6 Experiments

In this section, we evaluate our proposed methods summarized in Algorithm 1, namely IBW and MD, on different experiments on both toy and real data. In practice, in all our experiments, the gradients in Proposition 3.1 are computed with a Monte Carlo approximation involving $B$ samples from Gaussian distributions in the components. Our other hyperparameters are the following: $N$ the number of components in the mixture; $\gamma$ the step-size; and we also vary the initial values of the means and covariances of the candidate mixture. Our code is available at `https://github.com/margueritetalamon/VI-MIG`.

We compare our methods with different competitors. The algorithm presented in Lambert et al. [2022], based on a Bures geometry (as IBW) but updating full covariance matrices in the mixture, is abbreviated as BW (see Appendix E.1 for more details on BW). Note that the latter has a computational complexity and memory scaling proportionally to $N(d + d^2)$ instead of $N(d+1)$ for our schemes IBW and MD. The algorithm of [Huix et al., 2024], considering mixtures of isotropic Gaussians with shared variance $\epsilon\mathrm{Id}$, that only updates the means with Eq. (GD) while keeping the variance $\epsilon$ fixed, is denoted GD. The "shared-variance" version of our schemes (where $\forall i = 1, \ldots, N$, we impose the constraint $\epsilon^i = \epsilon$, but the latter is optimized) are denoted IBW-s and MD-s. The natural gradient descent algorithm of Lin et al. [2019], adapted to isotropic Gaussians with fixed weights and given in Eq (14), is referred to as NGD. We also evaluate a Normalizing Flow (NF) implementation based on Real NVPs [Dinh et al., 2017]. We also used Hamiltonian Monte Carlo (HMC), an MCMC scheme (hence non parametric) and Automatic Differentiation Variational Inference (ADVI) Kucukelbir et al. [2017]. All the experimental details are provided in Appendix E.2.

**Gaussian-mixture target in two dimensions.** We first illustrate the behavior of our methods for a two-dimensional target $\pi$ that is a Gaussian mixture with 5 components. In Figure 1 (top), we evaluate our methods for $N = 1, 5, 10, 20$. We observe several interesting facts. Choosing a number of components $N = 20$ leads to the lowest final KL objective. This is in accordance with the fact that the approximation error with a mixture of isotropic Gaussians goes to zero as $N$ tends to infinity, see Section 5. The shared-variance versions of our algorithms (IBW-s and MD-s) did not perform as well as the original schemes we propose, namely IBW and MD. This demonstrates the benefit of optimizing each component's variance to better fit a given target's shape. In Appendix E.3, we visualize the optimized Gaussian components (represented as circles), highlighting the benefit of allowing each component to have its own variance. The NF method appears slower than ours without improving the approximation. We plot the approximated density with $N = 10$ for BW, IBW, GD and NF methods together with the target density in Figure 1 (bottom).

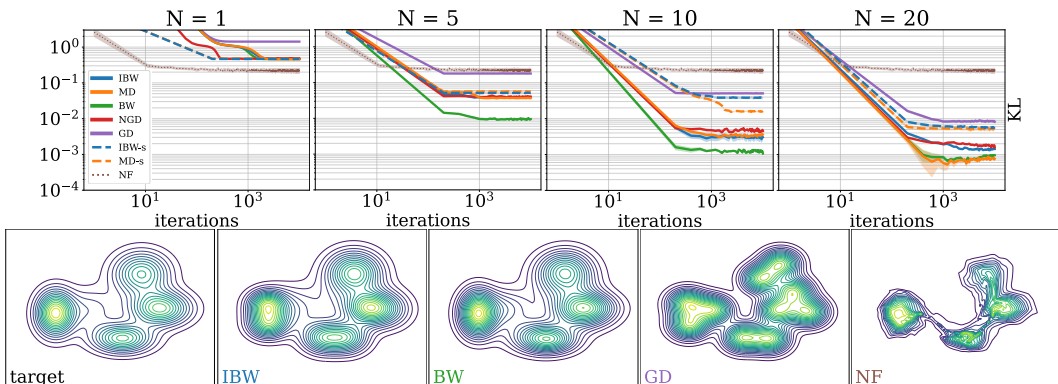

Figure 1: Illustration of convergence of Algorithm 1 for a two-dimensional target distribution.

We also evaluated these VI methods on alternative challenging two-dimensional target distributions, these results are deferred to the Appendix. More precisely, we test the performance of these methods on a Funnel distribution and heavy-tailed targets in Appendix E.3. Then we investigate how well our schemes on mixtures of isotropic Gaussians can fit challenging Gaussian mixtures (e.g. with highly unbalanced mode weights) in Appendix E.4.

**Gaussian-mixture target in high dimension.** We then consider a Gaussian mixture target with 10 components in dimension $d = 20$, and a variational mixture model with $N = 20$. In Figure 2, we plotted the marginals along each dimension together with the KL evolution for the schemes (BW, IBW, MD, NGD).

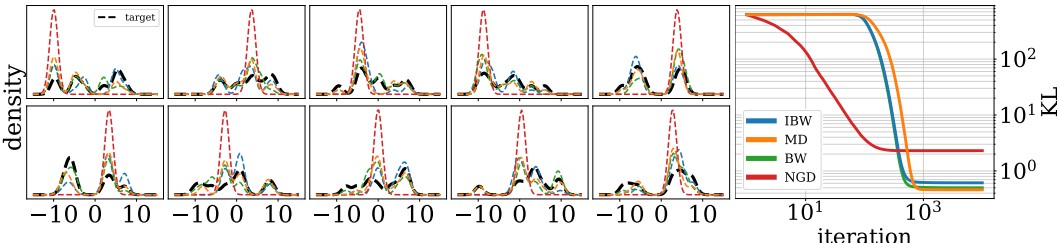

Figure 2: (First 10) Marginals (left) and KL objective (right) for MD, IBW, BW and NGD ($d = 20$).

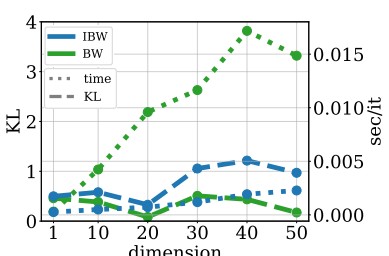

Figure 3: Time/KL evolution w.r.t. $d$.

We observe that both our schemes IBW and MD provide a good approximation, along with BW. In the mean update of NGD, the gradient is rescaled by $\epsilon$, which leads to a large step for a spread-out Gaussian. While this rescaling allows faster convergence, we observe that it makes the algorithm more sensitive to the initial conditions. Then, we compare in Figure 3 the time per iteration and the KL objective value for BW (that updates full covariances matrices), and our isotropic version (IBW) for a similar target over several dimensions, for $N = 15$. We note that IBW performs comparably to BW, while enjoying a faster time execution, and still with a cost in memory linear in the dimension instead of quadratic.

**Bayesian posteriors.** We evaluate our methods on two probabilistic inference tasks using classical datasets. The first one is Bayesian logistic regression (BLR) for two UCI datasets: `breast_cancer` (2 labels, $d = 30$) and `wine` (3 labels, $d = 39$). The second one aims to compute a Bayesian neural network (BNN) posterior on a regression task on the `boston` dataset using a single hidden layer neural network of 50 units ($d = 601$), and on the MNIST with a one layer neural network with 256 units ($d = 203530$). In each case, we assume a standard Gaussian prior on the parameters, and compute the posterior distribution given observations. More details are given in Appendix E.6 . Note that the first task leads to log-concave posteriors (in contrast to the previous mixture of Gaussians targets which are typically non log-concave) while the second typically leads to a multimodal one [Izmailov et al., 2021].

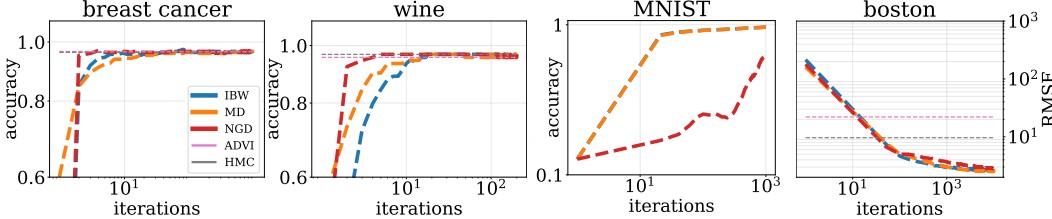

Figure 4: Bayesian logistic regression and BNN regression approximated by mixtures of Gaussians.

In Figure 4, we present the results of our algorithms IBW and MD for $N = 5$, on these Bayesian inference tasks, plotting the accuracy or Root Mean Squared Error (RMSE) score on the test set over iterations. The evolution of $\log$-likelihood and unnormalized KL (ELBO) are deferred to E.6. As additional baselines, we compared our results with HMC and ADVI methods (except on MNIST -where we did not manage to find a working set of hyperparam in such high dimensions), and only plotted the final samples results provided by `stan`. Other experiments, such as comparing the

performance of our methods with unimodal variational approximations, e.g. using $N = 1$ within our framework or popular Laplace approximations (e.g. Diagonal and K-FAC [Ritter et al., 2018]) on MNIST, are also deferred to E.6; they show the relevance of using several components to capture different modes, for such multimodal posteriors. Regarding MNIST, for scalability issues, in analogy with Blundell et al. [2015], we coded a mean-field version of our algorithm (see also Appendix E.6 for details) where each weight marginal is fitted by a Gaussian mixture with our Algorithm 1. We observed that we achieved the same order of performance for IBW and MD. We also performed the optimization with NGD updates but found out that if the means were not initialized within a very small ball (increasing the chances of missing some modes in the target), the variance $\epsilon$ estimated by NGD could become negative, which made the optimization difficult in practice. In contrast, our scheme guarantees the positivity of $\epsilon$ by construction.

# 7   Conclusion

Mixtures of isotropic Gaussians provide a simplified yet powerful tool in variational inference, balancing expressivity for multimodal target distributions with computational and memory efficiency. We presented two optimization schemes, that implement joint optimization on the means through gradient descent, and on the variances through adapted geometries for the space of variance matrices, such as the Bures or Von Neumann entropy ones, guaranteeing that they remain positive. Our numerical experiments validate their relevance for different types of target posterior distributions. Future work include establishing more theoretical guarantees regarding our schemes and mixtures of isotropic Gaussians. For instance, comparing the approximation error of full covariance mixtures versus isotropic ones, would be helpful to understand why we observe empirically a great computational cost gain for a very modest increase of the KL loss (e.g., IBW vs BW). Then, studying optimization guarantees for these schemes is of interest.

**Acknowledgements**   M P-T and AK thank Apple for their academic support in the form of a research funding. ML acknowledges support from the French Defence procurement agency (DGA). This project was also funded by the European Union (ERC, Optinfinite, 101201229). The views and opinions expressed are, however, of the author(s) only and do not necessarily reflect those of the European Union or the European Research Council Executive Agency. Neither the European Union nor the granting authority can be held responsible for them.

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

# A  Appendix

## A.1  Geometric structure of Isotropic Gaussians space

We prove here the stability of the isotropic Gaussian model along the Bures-Wasserstein geodesics. We recall that the Bures-Wasserstein space is the space of non-degenerate Gaussian distributions equipped with the Wasserstein-2 metric.

**Proposition A.1.** *The space of isotropic Gaussians equipped with the Bures-Wasserstein distance is a geodesically convex subset of the Bures-Wasserstein space, which is itself a geodesically convex subspace of the Wasserstein space $(\mathcal{P}_2(\mathbb{R}^d), \mathrm{W}_2)$.*

*Proof.* Let $p = \mathcal{N}(m_p, \epsilon_p \mathrm{I}_d)$ and $q = \mathcal{N}(m_q, \epsilon_q \mathrm{I}_d)$ be two isotropic Gaussian distributions. Since $p$ is absolutely continuous, the Wasserstein-2 geodesic, (which is also a Bures-Wasserstein geodesic) between $p$ and $q$ is given by the pushforward measure:

$$\mu_t = ((1-t)\mathrm{Id} + tT)_{\#}\, p, \quad t \in [0,1],$$

where $\mathrm{Id}$ is the identity map and $T$ is the optimal transport map between $p$ and $q$. Denoting $\Sigma_p = \epsilon_p \mathrm{Id}$ and $\Sigma_q = \epsilon_q \mathrm{Id}$, this optimal transport map $T$ is the affine map:

$$T(x) = m_q + A(x - m_p), \ \text{where } A = \Sigma_p^{-1/2}\left(\Sigma_p^{1/2}\Sigma_q\Sigma_p^{1/2}\right)^{1/2}\Sigma_p^{-1/2}.$$

$A$ is called the Bures map and satisfies $\Sigma_q = A\Sigma_p A$, which can be easily verified from the definition above. Since the map is linear, it preserves densities, ensuring that the transported measure $\mu_t$ remains Gaussian. The mean and covariance parameters evolve along the Bures-Wasserstein geodesic between $p$ and $q$ according to the equations:

$$m_t = (1-t)m_p + tm_q$$
$$\Sigma_t = ((1-t)\mathrm{I}_d + tA)\Sigma_p((1-t)\mathrm{I}_d + tA).$$

Since both $p$ and $q$ are isotropic, the transport map is:

$$T(x) = m_q + a(x - m_p), \ \text{where } a = \left(\frac{\epsilon_q}{\epsilon_p}\right)^{1/2}.$$

Then, the covariance $\Sigma_t$ evolves according to:

$$\Sigma_t = ((1-t)\mathrm{I}_d + ta\mathrm{I}_d)\epsilon_p((1-t)\mathrm{I}_d + ta\mathrm{I}_d) = ((1-t) + ta)^2\epsilon_p\mathrm{I}_d,$$

which is clearly isotropic. Hence, the interpolated distribution $\mu_t$ remains an isotropic Gaussian for all $t \in [0,1]$. Thus, the space of isotropic Gaussian distributions is geodesically convex in the Bures-Wasserstein space. Since the Bures-Wasserstein space is itself a geodesically convex subspace of the Wasserstein space, we complete the proof. □

## A.2  Proof of Theorem 2.1

### A.2.1  Background on Wasserstein gradient flows

Let $\mathcal{F} : \mathcal{P}_2(\mathbb{R}^d) \to \mathbb{R} \cup \{+\infty\}$ be a functional. We say that $\mathcal{F}$ is $\alpha$-convex along Wasserstein-2 geodesics if for any two probability measures $\mu_0, \mu_1 \in \mathcal{P}_2(\mathbb{R}^d)$ and any geodesic $\{\mu_t\}_{t \in [0,1]}$ in the Wasserstein-2 space connecting $\mu_0$ and $\mu_1$, we have

$$\mathcal{F}(\mu_t) \leq (1-t)\mathcal{F}(\mu_0) + t\mathcal{F}(\mu_1) - \frac{\alpha}{2}t(1-t)\,\mathrm{W}_2^2(\mu_0,\mu_1), \quad \forall t \in [0,1].$$

A *Wasserstein gradient flow* of $\mathcal{F}$ is a solution $(\mu_t)_{t \in (0,T)}$, $T > 0$, of the continuity equation

$$\frac{\partial \mu_t}{\partial t} + \nabla \cdot (\mu_t v_t) = 0$$

that holds in the distributional sense, where $v_t$ is a subgradient of $\mathcal{F}$ at $\mu_t$ [Ambrosio et al., 2008, Definition 10.1.1]. Among the possible processes $(v_t)_t$, one has a minimal $L^2(\mu_t)$ norm and is called the velocity field of $(\mu_t)_t$. In a Riemannian interpretation of the Wasserstein space [Otto, 2001],

this minimality condition can be characterized by $v_t$ belonging to the tangent space to $\mathcal{P}_2(\mathbb{R}^d)$ at $\mu_t$ denoted $T_{\mu_t}\mathcal{P}_2(\mathbb{R}^d)$, which is a subset of $L^2(\mu_t)$, the Hilbert space of square integrable functions with respect to $\mu_t$, whose inner product is denoted $\langle \cdot, \cdot \rangle_{\mu_t}$. The Wasserstein gradient is defined as this unique element, and is denoted $\nabla_{W_2}\mathcal{F}(\mu_t)$. In particular, if $\mu \in \mathcal{P}_2(\mathbb{R}^d)$ is absolutely continuous with respect to the Lebesgue measure, with density in $C^1(\mathbb{R}^d)$ and such that $\mathcal{F}(\mu) < \infty$, $\nabla_{W_2}\mathcal{F}(\mu)(x) = \nabla\mathcal{F}'(\mu)(x)$ for $\mu$-a.e. $x \in \mathbb{R}^d$ [Ambrosio et al., 2008, Lemma 10.4.1], where $\mathcal{F}'(\mu)$ denotes the first variation of $\mathcal{F}$ evaluated at $\mu$, i.e. (if it exists) the unique function $\mathcal{F}'(\mu) : \mathbb{R}^d \to \mathbb{R}$ s.t.

$$\lim_{h \to 0} \frac{1}{h}(\mathcal{F}(\mu + h\xi) - \mathcal{F}(\mu)) = \int \mathcal{F}'(\mu)(x)d\xi(x)$$

for all $\xi = \nu - \mu$, where $\nu \in \mathcal{P}_2(\mathbb{R}^d)$.

### A.2.2 Proof

The proof relies on tools on the Wasserstein geometry and calculus introduced above, and is a direct application of the one of [Lambert et al., 2022, Corollary 3], yet we state it for completeness. When $\nabla^2 V \succeq \alpha I_d$, $\mathcal{F} : \mu \mapsto \mathrm{KL}(\mu|\pi)$ is $\alpha$-convex on $\mathcal{P}_2(\mathbb{R}^d)$ [Villani, 2009, Theorem 17.15]. Let us denote $p^\star$ the minimum of this function and recall we denote BW the Bures-Wasserstein metric on the manifold $(\mathrm{IG}, \mathrm{BW})$. We consider the following gradient flow:

$$\frac{\partial p_t}{\partial t} = \mathrm{div}\,(p_t \nabla_{W_2}\mathcal{F}(p_t)) \text{ with the initial condition } p_0 = p_0.$$

We first want to show that the solution of this problem is unique. Let $(p_t)_t$ and $(q_t)_t$ be two solutions of the above gradient flow. Then, using differential calculus in the Wasserstein space and the chain rule, we have

$$\partial_t \mathrm{BW}^2(p_t, q_t) = 2\langle \log_{p_t}(q_t), \nabla_{W_2}\mathcal{F}(p_t)\rangle_{p_t} + 2\langle \log_{q_t}(p_t), \nabla_{W_2}\mathcal{F}(q_t)\rangle_{q_t},$$

where $\nabla_{W_2}\mathcal{F}(p)$ denotes the Wasserstein gradient at $p \in \mathcal{P}_2(\mathbb{R}^d)$ and $\log_{p_t}(q_t) = T - \mathrm{Id} \in L^2(p_t)$, where $T$ is the optimal transport map from $p_t$ to $q_t$. Moreover since $\mathcal{F}$ is $\alpha$ convex, we can write $\forall p, q \in \mathcal{P}_2(\mathbb{R}^d)$ :

$$\mathcal{F}(q) \geq \mathcal{F}(p) + \langle \nabla_{W_2}\mathcal{F}(p), \log_p(q)\rangle - \frac{\alpha}{2}\mathrm{BW}^2(p, q).$$

Thus we can write

$$\partial_t \mathrm{BW}^2(p_t, q_t) \leq -2\alpha\,\mathrm{BW}^2(p_t, q_t).$$

Hence, by Grönwall's lemma, we obtain

$$\mathrm{BW}^2(p_t, q_t) \leq \exp(-2\alpha t)\,\mathrm{BW}^2(p_0, q_0).$$

Since both $(p_t)_t$ and $(q_t)_t$ are solution of the gradient flow, $p_0 = q_0$ and it implies that $\forall t \in [0, 1], p_t = q_t$. This proves the uniqueness of the solution.

Moreover, if $\alpha > 0$, we can set $q_t = p^\star$ for all $t \geq 0$ to deduce exponential contraction of the gradient flow to the minimizer $p^\star$. Observe that by definition of the gradient flow, we have on the one hand that

$$\partial_t \mathcal{F}(p_t) = \langle \nabla_{W_2}\mathcal{F}(p_t), -\nabla_{W_2}\mathcal{F}(p_t)\rangle_{p_t} = -\|\nabla\mathcal{F}(p_t)\|^2_{p_t}. \tag{15}$$

On the other hand, if $\alpha > 0$, the convexity inequality and Young's inequality respectively, yield

$$0 = \mathcal{F}(p^\star) \geq \mathcal{F}(p) + \langle \nabla_{W_2}\mathcal{F}(p), \log_p(p^\star)\rangle_p + \frac{\alpha}{2}\mathrm{BW}^2(p, p^\star)$$

$$\geq \mathcal{F}(p) - \frac{1}{2\alpha}\|\nabla_{W_2}\mathcal{F}(p)\|^2_p - \frac{\alpha}{2}\underbrace{\left\|\log_p(p^\star)\right\|^2_p}_{=\mathrm{BW}^2(p, p^\star)} + \frac{\alpha}{2}\mathrm{BW}^2(p, p^\star)$$

and hence $\|\nabla_{W_2}\mathcal{F}(p)\|^2 \geq 2\alpha\,\mathcal{F}(p)$. Substituting this into Eq (15) and applying Grönwall's inequality again, we deduce

$$\mathcal{F}(p_t) \leq \exp(-2\alpha t)\,\mathcal{F}(p_0).$$

### A.3 JKO scheme for isotropic Gaussians

In this section, we derive the continuous equations for the isotropic Bures-Wasserstein gradient flow. Starting from the JKO scheme of Eq (3), we constrain the solution to lie in the space of isotropic Gaussians IG. We follow the same method than [Lambert et al., 2022, Appendix A].

Let us recall the JKO scheme [Jordan et al., 1998]. Starting from $p_0 = \mathcal{N}(m_0, \epsilon_0 \mathrm{I}_d)$ at time $k = 0$ we look for the solution $p = \mathcal{N}(m, \epsilon \mathrm{I}_d)$ of:

$$p_{k+1} = \operatorname*{arg\,min}_{p \in \mathrm{IG}} \left\{ \mathrm{KL}(p|\pi) + \frac{1}{2\gamma} \mathrm{BW}^2(p, p_k) \right\}. \tag{16}$$

The gradient of the KL with respect to the variance parameters is given by Eq (21):

$$\nabla_\epsilon \mathrm{KL}(p|\pi) = -\frac{d}{2\epsilon} - \frac{1}{2} \mathrm{Tr}\, \mathbb{E}_p[\nabla^2 \log \pi] = -\frac{d}{2\epsilon} + \frac{1}{2} \mathrm{Tr}\, \mathbb{E}_p[\nabla^2 V].$$

Then, for two isotropic Gaussian distributions $p = \mathcal{N}(m, \epsilon \mathrm{I}_d)$ and $p_k = \mathcal{N}(m_k, \epsilon_k \mathrm{I}_d)$ we have[2]

$$\mathrm{BW}^2(p, p_k) = \|m - m_k\|_2^2 + d\left(\epsilon + \epsilon_k - 2\sqrt{\epsilon \epsilon_k}\right),$$

and

$$\nabla_\epsilon \mathrm{BW}^2(p, p_k) = d\left(1 - \sqrt{\frac{\epsilon_k}{\epsilon}}\right).$$

Hence, the first order condition on $\epsilon$ of (16) yield:

$$d\left(1 - \sqrt{\frac{\epsilon_k}{\epsilon}}\right) = \frac{d\gamma}{\epsilon} - \gamma \mathrm{Tr}\, \mathbb{E}_p\left[\nabla^2 V\right] \Leftrightarrow \epsilon_k = \epsilon\left(1 - \frac{\gamma}{\epsilon} + \frac{\gamma}{d} \mathrm{Tr}\, \mathbb{E}_p\left[\nabla^2 V\right]\right)^2$$

$$\Leftrightarrow \epsilon_k = \epsilon\left(1 - \gamma\left(\frac{1}{\epsilon} - \frac{1}{d\epsilon} \mathbb{E}_p\left[(\cdot - m)^T \nabla V\right]\right)\right)^2. \tag{17}$$

Developing equation Eq (17) at first order in $\gamma$ we obtain:

$$\epsilon_k = \epsilon\left(1 - \frac{2\gamma}{\epsilon}\left(1 - \frac{1}{d}\mathbb{E}_p\left[(\cdot - m)^T \nabla V\right]\right) + \mathrm{O}(\gamma^2)\right)$$

$$\Leftrightarrow \frac{\epsilon - \epsilon_k}{\gamma} = 2 - \frac{2}{d}\mathbb{E}_p\left[(\cdot - m)^T \nabla V\right] + \mathrm{O}(\gamma).$$

Taking the limit $\gamma \to 0$ yields the differential equation:

$$\dot{\epsilon} = 2 - \frac{2}{d}\mathbb{E}_p\left[(\cdot - m)^T \nabla V\right].$$

For the first order condition on the mean parameter $m$ of (16), using the gradient of the KL w.r.t. the mean given Eq (20), we obtain:

$$\mathbb{E}_p\left[\nabla \log\left(\frac{p}{\pi}\right)\right] + \frac{1}{\gamma}(m - m_k) = 0. \tag{18}$$

Since $\mathbb{E}_p[\nabla \log p] = 0$ we obtain, at the limit $\gamma \to 0$:

$$\dot{m} = \mathbb{E}_p[\nabla \log \pi] = -\mathbb{E}_p[\nabla V].$$

Inspecting Eq (17–18), we observe that solving the JKO scheme yields an implicit discrete-time update, where the expectations are evaluated under the unknown distribution $p$. We now derive the explicit form of this update, starting from the formulation in Eq (4).

---

[2]Recall that for two general Gaussians $p = \mathcal{N}(m, \Sigma)$, $p_k = \mathcal{N}(m_k, \Sigma_k)$, we have $\mathrm{BW}^2(p, p_k) = \|m - m_k\|_2^2 + \mathrm{Tr}\left(\Sigma + \Sigma_k - 2\left(\Sigma^{1/2}\Sigma_k\Sigma^{1/2}\right)^{1/2}\right)$.

## A.4 Forward Euler scheme for isotropic Gaussians

**Derivations of the updates.** We now derive the updates given by the scheme Eq (4). The first order condition on $\epsilon$ is given by:

$$\frac{1}{2}d\left(1 - \sqrt{\frac{\epsilon_k}{\epsilon}}\right) = -\gamma\nabla_\epsilon F(m_k, \epsilon_k) \Leftrightarrow \epsilon = \epsilon_k\left(1 + \frac{2\gamma}{d}\nabla_\epsilon F(m_k, \epsilon_k)\right)^{-2}.$$

Using the Taylor expansion for small step $\gamma$ given by $(1 + x)^{-1} = 1 - x + o(x)$ we obtain at first order the explicit update:

$$\epsilon = (1 - \frac{2\gamma}{d}\nabla_\epsilon F(m_k, \epsilon_k))^2\epsilon_k.$$

The first-order condition on $m$ gives the explicit update:

$$m = m_k - \gamma\nabla_m F(m_k, \epsilon_k).$$

**Riemannian interpretation.** The latter scheme can be identified to Riemannian gradient descent, on the isotropic Bures-Wasserstein space, i.e. the space IG of isotropic Gaussians equipped with the Bures-Wasserstein metric, that we will denote iBW $= (\mathrm{IG}, \mathrm{BW})$. To achieve this, we first identify the local tangent space of the isotropic Bures-Wasserstein space. The direction of the tangent vector is computed by projecting the Wasserstein-2 gradient of the KL objective onto this tangent space (see Appendix A.2.1 for the definition). We then follow this projected gradient with a step size $\gamma$. We can then retract back to the isotropic Bures-Wasserstein manifold using an exponential map. We now detail this approach.

Let $\mathcal{F} : \mathcal{P}_2(\mathbb{R}^d) \to \mathbb{R}$ a functional. The isotropic Bures-Wasserstein Gradient of $\mathcal{F}$ at $p \in \mathrm{IG}$, denoted $\nabla_{\mathrm{iBW}}\mathcal{F}(p)$, is the projection of its Wasserstein gradient onto the tangent space to iBW at $p$. If $p = \mathcal{N}(m_p, \epsilon_p I_d)$, this tangent space writes:

$$T_p\mathrm{iBW}(\mathbb{R}^d) = \{x \mapsto a + s(x - m_p)|a \in \mathbb{R}^d, s \in \mathbb{R}\},$$

which can be identified with the pair $(a, s) \in \mathbb{R}^d \times \mathbb{R}$. Thus,

$$\nabla_{\mathrm{iBW}}\mathcal{F}(p) = \mathrm{proj}_{T_p\mathrm{iBW}(\mathbb{R}^d)}\nabla_{\mathrm{W}_2}\mathcal{F}(p) = \underset{w \in T_p\mathrm{iBW}(\mathbb{R}^d)}{\mathrm{argmin}}\|w - \nabla_{\mathrm{W}_2}\mathcal{F}(p)\|_p^2.$$

The first conditions in $a \in \mathbb{R}^d$ and $s \in \mathbb{R}$ of this problem yield:

$$a = \mathbb{E}_p\left[\nabla_{\mathrm{W}_2}\mathcal{F}(p)\right] \quad \text{and} \quad s = \frac{1}{d\epsilon_p}\mathbb{E}_p\left[(\cdot - m_p)^T\nabla_{\mathrm{W}_2}\mathcal{F}(p)\right].$$

Indeed,

$$\nabla_a \int \|a + s(x - m_p) - \nabla_{\mathrm{W}_2}\mathcal{F}(p)(x)\|^2 dp(x) = 0$$

$$\iff \int 2(a + s(x - m_p) - \nabla_{\mathrm{W}_2}\mathcal{F}(p)(x))dp(x) = 0$$

$$\iff a = \mathbb{E}_p\left[\nabla_{\mathrm{W}_2}\mathcal{F}(p)\right],$$

and

$$\nabla_s \int \|a + s(x - m_p) - \nabla_{W_2}\mathcal{F}(p)(x)\|^2 dp(x) = 0$$

$$\iff \int 2(x - m_p)^T(a + s(x - m_p) - \nabla_{\mathrm{W}_2}\mathcal{F}(p)(x))dp(x) = 0$$

$$\iff s\epsilon_p d - \int (x - m_p)^T\nabla_{\mathrm{W}_2}\mathcal{F}(p)(x)dp(x) = 0$$

$$\iff s = \frac{1}{d\epsilon_p}\mathbb{E}_p\left[(x - m_p)^T\nabla_{\mathrm{W}_2}\mathcal{F}(p)\right].$$

Back to our problem with $\mathcal{F}(p) = \text{KL}(p|\pi)$, using Eq (20) and (21) and that $\nabla_{W_2}\mathcal{F}(p) = \nabla \log\left(\frac{p}{\pi}\right)$ (see Appendix A.2.1) we have that

$$a = \nabla_m \text{KL}(p|\pi) = \nabla_m F(m_k, \epsilon_k) \quad \text{and} \quad s = \frac{2}{d}\nabla_\epsilon \text{KL}(p|\pi) = \frac{2}{d}\nabla_\epsilon F(m_k, \epsilon_k).$$

We can follow the direction of the gradient from the tangent space back to the isotropic Bures-Wasserstein manifold using an exponential map which is available in closed form

$$\exp_p(a, s) = \mathcal{N}\left(m_p + a, \ (1 + s)^2 \epsilon_p I_d\right), \tag{19}$$

where we have adapted the exponential map formula for the Bures-Wasserstein space [Lambert et al., 2022, Appendix B.3], written as $\exp_p(a, S) = \mathcal{N}\left(m_p + a, \ (S + I)\Sigma_p(S + I)\right)$, to the case of isotropic Gaussians. We can then construct a discrete update based as follow: we compute the iBW gradient of $\mathcal{F}(p) = \text{KL}(p|\pi)$ multiplied a step size $\gamma$, and map it back onto the manifold using this exponential map. Starting from $p_k = \mathcal{N}(m_k, \epsilon_k I_d)$, this discrete time update is:

$$p_{k+1} = \exp_{p_k}(-\gamma \nabla_{\text{iBW}}\mathcal{F}(p_k)) = \exp_{p_k}(-\gamma \nabla_m F(m_k, \epsilon_k), -\frac{2\gamma}{d}\nabla_\epsilon F(m_k, \epsilon_k)),$$

which gives the update on the mean and variance parameters:

$$m_{k+1} = m_k - \gamma \nabla_m F(m_k, \epsilon_k),$$
$$\epsilon_{k+1} = (1 - \frac{2\gamma}{d}\nabla_\epsilon F(m_k, \epsilon_k))^2 \epsilon_k.$$

## A.5 Background on Mirror Descent

Mirror descent is an optimization algorithm that was introduced to solve constrained convex problems [Nemirovskij and Yudin, 1983], and that uses in the optimization updates a cost (or "geometry") that is a Bregman divergence [Bregman, 1967], whose definition is given below.

**Definition A.2.** Let $\phi : \mathcal{X} \to \mathbb{R}$ a strictly convex and differentiable functional on a convex set $\mathcal{X}$, referred to as a Bregman potential. The $\phi$-Bregman divergence is defined for any $x, y \in \mathcal{X}$ by:

$$B_\phi(y|x) = \phi(y) - \phi(x) - \langle \nabla\phi(x), y - x \rangle.$$

Let $\gamma$ a fixed step-size and $G$ an objective function on $\mathcal{X}$. Mirror descent with $\phi$-Bregman divergence writes at each step $k \geq 0$ as:

$$x_{k+1} = \underset{x \in \mathcal{X}}{\text{argmin}} \ \langle \nabla G(x_k), x - x_k \rangle + \frac{1}{\gamma}B_\phi(x|x_k).$$

Writing the first order conditions of the problem above, mirror descent writes

$$x_{k+1} = \nabla\phi^*(\nabla\phi(x_k) - \gamma \nabla G(x_k)),$$

where $\phi^*(x) = \sup_{y \in \mathcal{X}}\langle y, x \rangle - \phi(x)$ is the convex conjugate of $\phi$, and $\nabla\phi^* = (\nabla\phi)^{-1}$. Note that, if $\mathcal{X}$ is a subset of $\mathbb{R}^d$ and that $\phi(x) = \frac{1}{2}\|x\|^2$, that $B_\phi(x, y) = \frac{1}{2}\|x - y\|^2$ and mirror descent coincides with gradient descent. Yet, this scheme is more general and is useful for constrained optimization, as if one chooses $\phi$ wisely, the inverse $\nabla\phi^*$ of the so-called "mirror map" $\nabla\phi$ maps the iterates into the domain of $\phi$.

## A.6 Entropic mirror descent updates

**Derivations of the updates (Gaussian case).** We now derive the updates given by the scheme Eq (2.3). The first order condition on $\epsilon$ is given by:

$$\frac{1}{2}d \log \frac{\epsilon}{\epsilon_k} = -\gamma \nabla_\epsilon F(m_k, \epsilon_k) \Leftrightarrow \epsilon = \epsilon_k \exp\left(-\frac{2\gamma}{d}\nabla_\epsilon F(m_k, \epsilon_k)\right).$$

The first order conditions on the means gives the same explicit update as Eq (4):

$$m = m_k - \gamma \nabla_m F(m_k, \epsilon_k)$$

*Remark* A.3. Note that Eq (2.3) involves a factor $1/2$ in front of the KL penalization term. Our motivation is that in Eq (4), the Bures distance $\mathrm{BW}^2(\epsilon\mathrm{Id}, \epsilon_k\mathrm{Id}) = d\left(\sqrt{\epsilon} - \sqrt{\epsilon_k}\right)^2$ is a distance between the square root of the matrices and thus its derivative w.r.t $\epsilon$ leads to a term implying only the square root of the matrices, or in our isotropic Gaussian setting the square root of the scale isotropic value: $\nabla_\epsilon \mathrm{BW}^2(\epsilon\mathrm{Id}, \epsilon_k\mathrm{Id}) = d\left(1 + \sqrt{\frac{\epsilon_k}{\epsilon}}\right)$. We find out it was judicious to have a derivative for the $\overline{\mathrm{KL}}$ implying also the square roots of the isotropic coefficient, i.e., $\nabla_\epsilon\overline{\mathrm{KL}}(\epsilon\mathrm{Id}|\epsilon_k\mathrm{Id}) = \frac{d}{2}\log\left(\frac{\epsilon}{\epsilon_k}\right) = d\log\left(\sqrt{\frac{\epsilon}{\epsilon_k}}\right)$.

**Mixture Case.** In the mixture of Gaussians case, we derive the following updates from Eq (4.3). The first order condition on $\epsilon^i$ for $i = 1, \cdots, N$ gives:

$$\frac{1}{2N}d\log\frac{\epsilon^j}{\epsilon_k^i} = -\gamma\nabla_{\epsilon^i}F([m_k^j, \epsilon_k^j]_{j=1}^N) \Leftrightarrow \epsilon^i = \epsilon_k^i\exp\left(-\frac{2N\gamma}{d}\nabla_{\epsilon^i}F([m_k^j, \epsilon_k^j]_{j=1}^N)\right),$$

and for the means:

$$m^i = m_k^i - N\gamma\nabla_{m^i}F([m_k^j, \epsilon_k^j]_{j=1}^N).$$

## A.7 Proof of Theorem 3.1

In this section, we aim to compute the gradients of the $\mathrm{KL}(\cdot|\pi)$ objective function defined Eq (7). To achieve this, a fundamental tool are Stein's identities [Stein, 1981] which relate the derivatives with respect to the parameters to the derivatives of the integrand.

### A.7.1 Stein's identities for isotropic Gaussians

**Lemma A.4.** *Let $p$ be an isotropic Gaussian $\mathcal{N}(m, \epsilon\mathrm{I}_d)$ and $p(x)$ its density for $x \in \mathbb{R}^d$. Assume that $\pi \in C^1(\mathbb{R}^d)$ and $\lim_{\|x\|\to\infty} p(x)\log\pi(x) = 0$. We have*

$$\nabla_m\mathrm{KL}(p|\pi) = \mathbb{E}_p\left[\nabla\log\left(\frac{p}{\pi}\right)\right] \tag{20}$$

$$\nabla_\epsilon\mathrm{KL}(p|\pi) = \frac{1}{2\epsilon}\mathbb{E}_p[(\cdot - m)^T\nabla\log\left(\frac{p}{\pi}\right)]. \tag{21}$$

*Note that the latter equation does not require to compute the Hessian (of dimension $d \times d$), which can be computationally more efficient than an alternative formula given in Eq (23).*

*Proof.* Recall that $p(x; \theta) = (2\pi\epsilon)^{-d/2}\exp\left(-\frac{\|x-m\|^2}{2\epsilon}\right)$ and $\mathrm{KL}(p|\pi) = \mathbb{E}_p\left[\log\left(\frac{p}{\pi}\right)\right]$. First, note that for $\theta = (m, \epsilon)$:

$$\nabla_\theta\int\log p(x; \theta)p(x; \theta)dx = \int\log p(x; \theta)\nabla_\theta p(x; \theta)dx,$$

which comes from the fact that the expectation of the score function is null, namely $\int\nabla_\theta\log p(x; \theta)p(x; \theta)dx = 0$. Hence,

$$\nabla_m\mathbb{E}_p\left[\log\left(\frac{p}{\pi}\right)\right] = \int\log\left(\frac{p}{\pi}(x)\right)\nabla_m p(x)dx = \int\nabla_x\log\left(\frac{p}{\pi}(x)\right)p(x)dx,$$

where we used $\nabla_m p(x) = -\nabla_x p(x)$ and an integration by parts. We now compute the gradient of $p$ with respect to $\epsilon$:

$$\nabla_\epsilon p(x) = \frac{1}{2}p(x)\left(\frac{1}{\epsilon^2}\|x - m\|^2 - \frac{d}{\epsilon}\right)$$

$$= \frac{1}{2}\mathrm{Tr}\left[p(x)\left(\frac{1}{\epsilon^2}(x - m)(x - m)^T - \frac{1}{\epsilon}\mathrm{I}_d\right)\right] = \frac{1}{2}\mathrm{Tr}\,\nabla_x^2 p(x). \tag{22}$$

Then, we have, using an integration by parts:

$$\nabla_\epsilon\mathbb{E}_p\left[\log\left(\frac{p}{\pi}\right)\right] = \frac{1}{2}\mathrm{Tr}\int\log\left(\frac{p}{\pi}\right)(x)\nabla_x^2 p(x)dx$$

$$= -\frac{1}{2}\mathrm{Tr}\int\nabla_x p(x)\nabla_x\log\left(\frac{p}{\pi}(x)\right)^T dx = \frac{1}{2\epsilon}\mathbb{E}_p[(\cdot - m)^T\nabla\log\left(\frac{p}{\pi}\right)].$$

Note that applying twice an integration by parts would yield another formula:

$$\nabla_\epsilon \mathbb{E}_p \left[ \log \left( \frac{p}{\pi} \right) \right] = \frac{1}{2} \operatorname{Tr} \mathbb{E}_p \left[ \nabla^2 \log \left( \frac{p}{\pi} \right) \right] = -\frac{d}{2\epsilon} - \frac{1}{2} \operatorname{Tr} \mathbb{E}_p \left[ \nabla^2 \log \pi \right]. \tag{23}$$

$\square$

### A.7.2 Stein's identities for mixtures of isotropic Gaussians

The application of the previous results for a mixture of isotropic Gaussians is straightforward. Let $\mu = \frac{1}{N} \sum_{i=1}^N \delta_{m^i}$, and recall we denote the associated isotropic Gaussian mixture as $k_\epsilon \otimes \mu = \frac{1}{N} \sum_{i=1}^N k_{\epsilon^i} \star \delta_{m^i} = \frac{1}{N} \sum_{i=1}^N \mathcal{N}(m^i, \epsilon^i \mathrm{I}_d)$ where $k_\epsilon(x) = (2\pi\epsilon)^{-d/2} \exp\left(-\|x\|^2/(2\epsilon)\right)$ is the Gaussian kernel.

Using the fact that the expectation of the score function is null and that $\nabla_{m^i} k_\epsilon \otimes \mu = \frac{1}{N} \nabla_{m^i} k_{\epsilon^i} \star \delta_{m^i}$, the gradient of the KL with respect to the mean parameter $m^i$ is then given by:

$$\nabla_{m^i} \operatorname{KL} \left( k_\epsilon \otimes \mu \,\middle|\, \pi \right) = \nabla_{m^i} \mathbb{E}_{k_\epsilon \otimes \mu} \left[ \ln \left( \frac{k_\epsilon \otimes \mu}{\pi} \right) \right] = \frac{1}{N} \mathbb{E}_{k_{\epsilon^i} \star \delta_{m^i}} \left[ \nabla \ln \left( \frac{k_\epsilon \otimes \mu}{\pi} \right) \right].$$

With the same arguments and using Eq (22), the gradient with respect to the variance parameter $\epsilon^i$ is:

$$\nabla_{\epsilon^i} \operatorname{KL} \left( k_\epsilon \otimes \mu \,\middle|\, \pi \right) = \nabla_{\epsilon^i} \mathbb{E}_{k_\epsilon \otimes \mu} \left[ \ln \left( \frac{k_\epsilon \otimes \mu}{\pi} \right) \right] = \frac{1}{2N\epsilon^i} \mathbb{E}_{k_{\epsilon^i} \star \delta_{m^i}} \left[ (\cdot - m^i)^T \nabla \ln \left( \frac{k_\epsilon \otimes \mu}{\pi} \right) \right].$$

Equivalently, note that for the latter gradient, using twice an integration by parts would yield the formula

$$\nabla_{\epsilon^i} \operatorname{KL} \left( k_\epsilon \otimes \mu \,\middle|\, \pi \right) = \frac{1}{2N} \operatorname{Tr} \mathbb{E}_{k_{\epsilon^i} \star \delta_{m^i}} \left[ \nabla^2 \ln \left( \frac{k_\epsilon \otimes \mu}{\pi} \right) \right].$$

## B   Bures-Wasserstein gradient flow for mixtures

In this section we consider the Bures-Wasserstein gradient flow for mixtures proposed in Lambert et al. [2022] in the particular case of isotropic covariance matrices.

### B.1   Additional discussion

Starting from a mixture of isotropic Gaussians $\nu_0 \in \mathcal{C}^N$ at $k = 0$, the JKO scheme where we constrain the distribution to be such a mixture writes recursively at subsequent step $k + 1$:

$$\hat{\nu}_{k+1} = \arg\min_{\nu \in \mathcal{C}^N} \left\{ \operatorname{KL}(\nu|\pi) + \frac{1}{2\gamma} \operatorname{W}_2^2(\nu, \nu_k) \right\}. \tag{24}$$

Unfortunately, a closed-form solution for this scheme is not available, as the Wasserstein distance between mixtures is not tractable. Regarding the geometry of the space of mixtures of Gaussians, Lambert et al. [2022] considered a mixing measure with infinitely many components $\mu \in \mathcal{P}(\Theta)$, which can be identified with a Gaussian mixture of the form $\int \mathcal{N}_\theta \, dp(\theta)$, where $\mathcal{N}_\theta$ is a Gaussian distribution with parameters $\theta \in \Theta$. Transport maps can be defined for this model, and gradient flows can subsequently be computed [Lambert et al., 2022, Theorem 5]. However, this model is highly *overparameterized*. For instance, as illustrated in Chewi et al. [2024], a single standard Gaussian in $\mathbb{R}$ can be represented by infinitely many mixing measures of the form $\mathcal{N}(0, \mathrm{Id}) = \int \mathcal{N}(x, \tau \mathrm{Id}) \, dp(x)$, where $p = \mathcal{N}(0, (1 - \tau)I)$ for any $\tau \in [0, 1]$.

An alternative solution is to constrain the mixing model by considering a fixed and finite number of components with uniform weights as we do in this work. This resolves the identifiability issue: in particular, two co-localized components with weight $w$ cannot be confused with a single component of weight $2w$, since all weights are equal. See [Delon and Desolneux, 2020, Proposition 2] for further details on identifiability for the mixture model. Moreover discrete measures with the same number of atoms are stable along Wasserstein-2 geodesics. This property still holds if we consider discrete measures on the isotropic Bures-Wasserstein space $\hat{p} = \frac{1}{N} \sum_{i=1}^N \delta_{(m^i, \epsilon^i)}$. We can therefore consider

discrete mixing measures with uniform weights as stable and identifiable representatives of Gaussian mixtures.

Following Section 4.2 we can reconsider the JKO scheme 24 where we replace the $W_2$ distance with the $W_{bw}$ on mixing measures:

$$\min_{m^i, \epsilon^i} \left\{ \mathrm{KL}(\nu|\pi) + \frac{1}{2\gamma} W_{bw}^2(\hat{p}, \hat{p}_k) \right\},$$

which, at the limit $\gamma \to 0$, is equivalent to:

$$\min_{m^i, \epsilon^i} \left\{ \mathrm{KL}(\nu|\pi) + \frac{1}{2N\gamma} \sum_{i=1}^N \mathrm{BW}^2(\mathcal{N}(m^i, \epsilon^i \mathrm{Id}), \mathcal{N}(m_k^i, \epsilon_k^i \mathrm{Id})) \right\}. \tag{25}$$

The problem in Eq. (25) corresponds to the discrete scheme introduced for the full covariance case $\Sigma^i = \epsilon^i I$ in [Lambert et al., 2022, Appendix B], without further justification. This scheme can therefore be reinterpreted as a gradient flow on the space of discrete mixing measures.

## B.2 Flow of mixtures of isotropic Gaussian

We now solve the problem (25) when $\nu$ is a mixture of isotropic Gaussians. We recall our notation for an isotropic Gaussian mixture: $\nu = k_\epsilon \otimes \mu := \frac{1}{N} \sum_{i=1}^N \mathcal{N}(m^i, \epsilon^i \mathrm{I}_d)$ where $k_{\epsilon^i} \star \delta_{m^i} := \mathcal{N}(m^i, \epsilon^i \mathrm{I}_d)$, as well as our loss function:

$$F([m^i, \epsilon^i]_{i=1}^N) := \mathrm{KL}\left( \frac{1}{N} \sum_{i=1}^N \mathcal{N}(m^i, \epsilon^i \mathrm{I}_d) \,\middle|\, \pi \right) = \mathrm{KL}(k_\epsilon \otimes \mu | \pi).$$

The derivative of this loss w.r.t. the parameters of the Gaussian mixture are given in Appendix A.7.2 and we report them here:

$$\nabla_{m^i} F = \frac{1}{N} \mathbb{E}_{k_{\epsilon^i} \star \delta_{m^i}} \left[ \nabla \ln \left( \frac{k_\epsilon \otimes \mu}{\pi} \right) \right], \tag{26}$$

$$\nabla_{\epsilon^i} F = \frac{1}{2N\epsilon^i} \mathbb{E}_{k_{\epsilon^i} \star \delta_{m^i}} \left[ (\cdot - m^i)^T \nabla \ln \left( \frac{k_\epsilon \otimes \mu}{\pi} \right) \right]. \tag{27}$$

Using the derivative of the Bures-Wasserstein distance computed in Appendix A.3, we obtain the first order condition w.r.t. $\epsilon^i$ for the JKO-like scheme (25):

$$d \left( 1 - \sqrt{\frac{\epsilon_k^i}{\epsilon^i}} \right) = -2N\gamma \nabla_{\epsilon^i} F \quad \Rightarrow \quad \epsilon_k^i = \left( 1 + \frac{2N\gamma}{d} \nabla_{\epsilon^i} F \right)^2 \epsilon^i,$$

and following Appendix A.3, as $\gamma \to 0$ we obtain the flow:

$$\dot{\epsilon}^i = -\frac{2}{d} \mathbb{E}_{k_{\epsilon^i} \star \delta_{m^i}} \left[ (\cdot - m^i)^T \nabla \ln \left( \frac{k_\epsilon \otimes \mu}{\pi} \right) \right].$$

On the other hand, first order condition on the mean gives the implicit update:

$$m^i = m_k^i - N\gamma \nabla_{m^i} F,$$

and at the limit $\gamma \to 0$ we obtain the flow:

$$\dot{m}^i = -\mathbb{E}_{k_{\epsilon^i} \star \delta_{m^i}} \left[ \nabla \ln \left( \frac{k_\epsilon \otimes \mu}{\pi} \right) \right].$$

## B.3 Discrete update and Gaussian particles

**Derivations of the updates.** For the scheme given in Eq (11), we obtain the following first-order conditions on the parameters for $i = 1, \cdots, N$:

$$\frac{1}{2N} d \left( 1 - \sqrt{\frac{\epsilon_k^i}{\epsilon^i}} \right) = -\gamma \nabla_{\epsilon^i} F([m_k^j, \epsilon_k^j]_{j=1}^N) \Leftrightarrow \epsilon^i = \epsilon_k^i \left( 1 + \frac{2N\gamma}{d} \nabla_{\epsilon^i} F([m_k^j, \epsilon_k^j]_{j=1}^N) \right)^{-2},$$

and using a Taylor expansion for small step size $\gamma$, $(1 + x)^{-1} = 1 - x + o(x)$ we obtain:

$$\epsilon^i = \epsilon^i_k \left( 1 - \frac{2N\gamma}{d} \nabla_{\epsilon^i} F([m^j_k, \epsilon^j_k]^N_{j=1}) \right)^2 . \tag{28}$$

The first-order condition on $m^i$ gives the explicit update:

$$m^i = m^i_k - N\gamma \nabla_{m^i} F([m^j_k, \epsilon^j_k]^N_{j=1}). \tag{29}$$

**Riemannian interpretation.** We now show that this update has also a Riemannian interpretation, extending our geometric analysis from Appendix A.4 to the case of mixtures. We can compute the isotropic Bures-Wasserstein gradient for each Gaussian component, by projecting the Wasserstein-2 gradient of the KL objective to the IBW-tangent space of each component. Each Gaussian particle (component) follows its own trajectory ruled by $\nabla_{\mathrm{iBW}^i} \mathcal{F}$. Adopting this point of view, we obtain the following system of updates for $i = 1, \ldots, N$:

$$p^i_{k+1} = \exp_{p^i_k}(-\gamma \nabla_{\mathrm{iBW}^i} \mathcal{F}(\nu_k)),$$

where $\exp_{p^i_k}$ is the exponential map from the iBW tangent space at $p^i_k$ defined in Appendix A.4. We then get:

$$\nabla_{\mathrm{iBW}^i} \mathcal{F}(\nu) = \mathrm{proj}_{T_{p^i} \mathrm{iBW}(\mathbb{R}^d)} \nabla_{\mathrm{W}_2} \mathcal{F}(\nu) = \underset{w \in T_{p^i} \mathrm{iBW}(\mathbb{R}^d)}{\mathrm{argmin}} \|w - \nabla_{\mathrm{W}_2} \mathcal{F}(\nu)\|^2_{p^i},$$

with $w = (a, s) \in \mathbb{R}^d \times \mathbb{R}$. Together with (26,27) it gives:

$$a = \mathbb{E}_{p^i} [\nabla_{\mathrm{W}_2} \mathcal{F}(\nu)] = N \nabla_{m^i} F([m^j, \epsilon^j]^N_{j=1}),$$

$$s = \frac{1}{d\epsilon^i} \mathbb{E}_{p^i} [(\cdot - m^i)^T \nabla_{\mathrm{W}_2} \mathcal{F}(\nu)] = \frac{2N}{d} \nabla_{\epsilon^i} F([m^j, \epsilon^j]^N_{j=1}).$$

Using Eq (19), we obtain the discrete updates Eq (28)-(29).

## B.4 Background on Wasserstein distances for Gaussian mixtures [Delon and Desolneux, 2020]

In Delon and Desolneux [2020], the authors introduce the $\mathrm{MW}_2$ distance as a Wasserstein distance between Gaussian mixtures, where the transport plan is itself constrained to be a Gaussian mixture (with any number of components). We denote by $GMM$ the latter space.

Namely, let $p_0, p_1$ being two general Gaussians mixtures $p_0 = \sum_{i=1}^{K_0} \pi^i_0 p^i_0$ and $p_1 = \sum_{i=1}^{K_1} \pi^i_1 p^i_1$, the $\mathrm{MW}_2$ distance is defined as:

$$\mathrm{MW}^2_2(p_0, p_1) = \min_{\gamma \in \mathcal{S}(p_0, p_1) \cap GMM} \int \|x_1 - x_2\|^2 d\gamma(x_1, x_2) \geq W^2_2(p_0, p_1),$$

which is an upper bound to the true Wasserstein distance since the transport plan is constrained.

It has been shown in [Delon and Desolneux, 2020, Proposition 4] that this distance is also equal to:

$$\mathrm{MW}^2_2(p_0, p_1) = \min_{W \in \mathcal{S}(\pi_0, \pi_1)} \sum_{i=1}^{K_0} \sum_{j=1}^{K_1} W_{i,j} \, \mathrm{BW}^2(\mathcal{N}(m_i, \Sigma_i), \mathcal{N}(m_j, \Sigma_j)),$$

where $\mathcal{S}(\pi_0, \pi_1)$ is the set of coupling matrices between the vector of weights $\pi_0$ and $\pi_1$ of the two mixtures defined by $\mathcal{S}(\pi_0, \pi_1) = \{W \in \mathcal{M}_{K_0 \times K_1}(\mathbb{R}^+) | \forall i, \ \sum_j W_{ij} = \pi^i_0; \forall j, \ \sum_i W_{ij} = \pi^j_1\}$, where we note $\mathcal{M}_{n \times p}(\mathbb{R}^+)$ the set of matrices of size $n \times p$ with positive values. Moreover, Delon and Desolneux [2020] showed that the optimal transport plan takes the form:

$$\gamma(x, y) = \sum_{i=1}^{K_0} \sum_{j=1}^{K_1} W^*_{i,j} p^i_0(x) \delta_{y = T^{BW}_{i,j}(x)}(y),$$

where $W^*$ is the optimal coupling matrix and $T^{BW}_{i,j}$ is the BW transport map from Gaussian component $i$ to Gaussian component $j$. The transport plan $\gamma$ is a GMM with at most $K_0 K_1$ Gaussian components which are degenerated.

**Mixture model with fixed number of components.**   We now consider the case of mixture of exactly $N$ Gaussians with equal, fixed weights:

$$p = \frac{1}{N} \sum_{i=1}^{N} \mathcal{N}(m^i, \Sigma^i) = \frac{1}{N} \sum_{i=1}^{N} p^i,$$

where $p^i$ is the $i^{th}$ Gaussian component of the mixture.

The $\mathrm{MW}_2$ distance takes an even simpler expression when considering a distance between two mixtures $p_0, p_1$ of exactly $N$ Gaussians with equal, fixed weights. If we note $\mathfrak{S}_N$ the set of permutations over $\{1, \ldots, N\}$, we have:

$$\mathrm{MW}_2^2(p_0, p_1) = \min_{\sigma \in \mathfrak{S}_N} \frac{1}{N} \sum_{i=1}^{N} \mathrm{BW}^2(\mathcal{N}(m_i, \Sigma_i), \mathcal{N}(m_{\sigma(i)}, \Sigma_{\sigma(i)})) = W_{bw}^2(\hat{p}_0, \hat{p}_1),$$

where $W_{bw}$ is the Wasserstein distance between mixing measures defined in Section 4.2. The lower and upper bounds for $\mathrm{MW}_2^2$ given in [Delon and Desolneux, 2020, Proposition 6] then transfer to $W_{bw}$ and give:

$$\mathrm{W}_2(p_0, p_1) \leq W_{bw}(\hat{p}, \hat{p}_k) \leq \mathrm{W}_2(p_0, p_1) + \sqrt{\frac{2}{N} \sum_{k=1}^{N} \mathrm{Tr}\Sigma_0^k} + \sqrt{\frac{2}{N} \sum_{k=1}^{N} \mathrm{Tr}\Sigma_1^k}.$$

The last term simplifies for isotropic Gaussians and we finally obtain:

$$0 \leq W_{bw}^2(\hat{p}, \hat{p}_k) - \mathrm{W}_2^2(\nu, \nu_k) \leq 2\sqrt{2d\epsilon^*}, \tag{30}$$

where $\epsilon^*$ is the maximal variance of the mixtures $\nu, \nu_k$. When $\epsilon^* \to 0$, such that the Gaussian mixture degenerates into an empirical measure, the two distance matches.

**Geodesics on mixtures**   Finally, when considering mixtures with $N$ equal weights, the transport plan has exactly $N$ components and can be written:

$$\gamma(x, y) = \frac{1}{N} \sum_{i=1}^{N} p_0^i(x) \delta_{y=T_{i,\sigma^*(i)}^{BW}(x)}(y),$$

such that the mixture model with exactly $N$ components is stable along the geodesics transported by this plan. Indeed, the intermediate measure between two GMM $p_0$ and $p_1$ is given by the formula for $t \in [0, 1]$:

$$\mu_t = P_t \# \gamma \text{ where } P_t(x) = (1 - t)x + ty.$$

Applying this to our specific case, we obtain:

$$p_t = \frac{1}{N} \sum_{i=1}^{N} ((1 - t)\mathrm{Id} + tT_{i,\sigma^*(i)}^{BW}) \# p_0^i,$$

where $p_t$ has exactly $N$ components, proving that our GMM structure with $N$ components is stable along the geodesics.

We may wonder if a transport map exists in our simpler framework of mixtures with a fixed number of components and equal weights. Unfortunately, this is not the case, as illustrated in Figure 5, where we observe that the map between the two mixtures is not bijective and cannot be represented by a function $T(x)$.

## C   Natural gradient descent updates

In this section, we give more details on Natural gradient descent on IG, which corresponds to the algorithm proposed by Lin et al. [2019]. NGD adapts standard gradient descent to the geometry of the parameter space, by preconditioning the Euclidean gradient with the inverse Fisher information matrix.

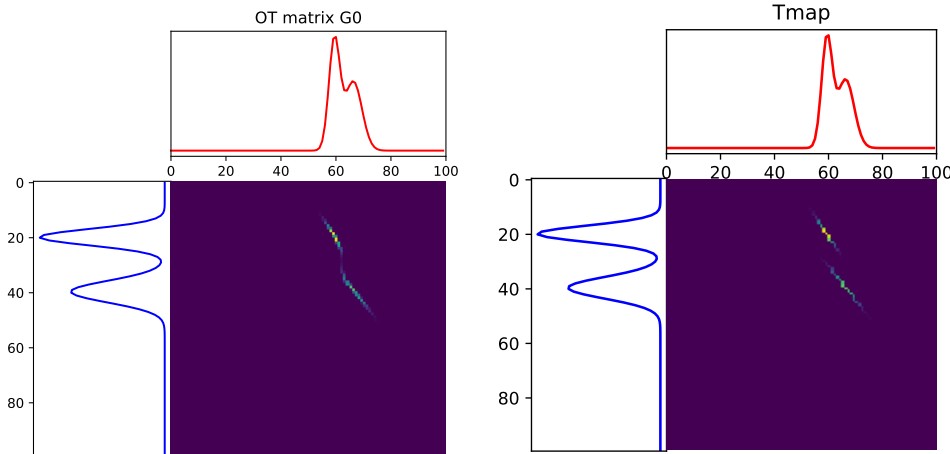

Figure 5: Optimal transport plan between two mixtures of two Gaussians with equal weights $\frac{1}{2}$. On the left is the Wasserstein distance $W_2$ case, where the optimal transport plan is not constrained. On the right is the $MW_2$ case, where the optimal transport plan is constrained to be a mixture of Gaussians. In the $W_2$ case, there exists a bijective map. We don't have such a bijective map for the $MW_2$ case. Indeed, in the right figure, some points have two images. These figures are generated using the Python Optimal Transport library (https://pythonot.github.io/).

**Exponential Family**   An isotropic Gaussian $p = \mathcal{N}(m, \epsilon I_d)$ belongs to the exponential family. Its density, in the canonical form, writes:

$$p(x; \eta) = (2\pi)^{-d/2} \exp\left(\langle \eta, S(x)\rangle - A(\eta)\right)$$

with the natural parameter $\eta = \begin{pmatrix} \eta_1 & \eta_2 \end{pmatrix}^\top = \begin{pmatrix} \frac{m}{\epsilon} & -\frac{1}{2\epsilon} \end{pmatrix}^\top$, the sufficient statistics $S(x) = \begin{pmatrix} x & \|x\|^2 \end{pmatrix}^\top$ and the log partition function $A(\eta) = -\frac{\|\eta_1\|^2}{2\eta_2} - \frac{d}{2}\log(-2\eta_2)$. It follows that $\nabla A(\eta) = \mathbb{E}_{p(\cdot|\eta)}[S(x)] := \beta$ and $\nabla^2 A(\eta) = I(\eta)$ with $\beta$ and $I(\eta)$ being respectively the mean parameter and the Fisher information matrix. Also $\nabla A^{-1} = \nabla A^*$ where $A^*(y) = \sup_\eta \langle \eta, y\rangle - A(\eta)$ is the Legendre transform of $A$.

**NGD as Mirror Descent [Raskutti and Mukherjee, 2015]**   Let $\eta \mapsto f(\eta)$ be the optimization objective in the natural-parameter space, and define the corresponding mean-space objective $\beta \mapsto \mathbf{f}(\beta) = f(\nabla A^*(\beta))$.

The natural-gradient step on $f$ of size $\gamma$ is

$$\eta_{k+1} = \eta_k - \gamma I(\eta_k)^{-1}\nabla f(\eta_k) = \eta_k - \gamma\left(\nabla^2 A(\eta_k)\right)^{-1}\nabla f(\eta_k). \tag{31}$$

On the other hand, mirror descent on $\mathbf{f}$ with the Bregman potential $\phi(\beta) = A^*(\beta)$ updates writes

$$\nabla A^*(\beta_{k+1}) = \nabla A^*(\beta_k) - \gamma\nabla_\beta \mathbf{f}(\beta_k). \tag{32}$$

By the chain rule,

$$\nabla_\beta \mathbf{f}(\beta_k) = \nabla^2 A^*(\beta_k)\nabla f(\eta_k) = \left(\nabla^2 A(\eta_k)\right)^{-1}\nabla f(\eta_k),$$

substituting this into the mirror descent update recovers exactly the NGD step above. The corresponding optimization scheme is:

$$\operatorname*{argmin}_\beta \langle \nabla_\beta \mathbf{f}(\beta_k), \beta - \beta_k\rangle + \frac{1}{\gamma}B_{A^\star}(\beta|\beta_k),$$

where the geometry is induced by the Bregman divergence $B_{A^*}$ generated by the Legendre transform $A^*$ of the partition function. This divergence is equal to the KL divergence between two isotropic Gaussians:

$$\mathrm{KL}(\mathcal{N}(m, \epsilon\mathrm{Id}), \mathcal{N}(m_k|\epsilon_k\mathrm{Id})) = \frac{1}{2}\left(d \cdot \frac{\epsilon}{\epsilon_k} + \frac{\|m - m_k\|^2}{\epsilon_k} - d + d\log\frac{\epsilon_k}{\epsilon}\right). \quad (33)$$

**Explicit Updates in** $(m, \epsilon)$   Define $F(m, \epsilon) = \mathbf{f}(m, \|m\|^2 + d\epsilon)$, and let $\nabla\mathbf{f}(\beta_k) = \begin{pmatrix} g_1 & g_2 \end{pmatrix}^\top$, with $\nabla_m F = g_1 + 2mg_2, \nabla_\epsilon F = dg_2$. Equivalently (31) and (32) give

$$\begin{pmatrix} \frac{m_{k+1}}{\epsilon_{k+1}} \\ -\frac{1}{2\epsilon_{k+1}} \end{pmatrix} = \begin{pmatrix} \frac{m_k}{\epsilon_k} \\ -\frac{1}{2\epsilon_k} \end{pmatrix} - \gamma\nabla\mathbf{f}(\beta_k) \Leftrightarrow \begin{pmatrix} m_{k+1} \\ \frac{1}{\epsilon_{k+1}} \end{pmatrix} = \begin{pmatrix} \frac{m_k}{\epsilon_k}\epsilon_{k+1} - \gamma g_1\epsilon_{k+1} \\ \frac{1}{\epsilon_k} + 2\gamma g_2 \end{pmatrix}$$

The variance update writes

$$\frac{1}{\epsilon_{k+1}} = \frac{1}{\epsilon_k} + \frac{2\gamma}{d}\nabla_\epsilon F(m_k, \epsilon_k). \quad (34)$$

and

$$
\begin{aligned}
m_{k+1} &= \left(\frac{m_k}{\epsilon_k} - \gamma g_1\right)\epsilon_{k+1} \\
&= \left(\frac{m_k}{\epsilon_k} - \gamma g_1\right)\frac{\epsilon_k}{1 + 2\epsilon_k\gamma g_2} \\
&= \left(\frac{m_k}{\epsilon_k} - \gamma\left[\nabla_m F(m_k, \epsilon_k) - 2m_k g_2\right]\right)\frac{\epsilon_k}{1 + 2\epsilon_k\gamma g_2} \\
&= \left(m_k\left[\frac{1}{\epsilon_k} + 2\gamma g_2\right] - \gamma\nabla_m F(m_k, \epsilon_k)\right)\frac{\epsilon_k}{1 + 2\epsilon_k\gamma g_2} \\
&= m_k - \gamma\nabla_m F(m_k, \epsilon_k)\frac{\epsilon_k}{1 + 2\epsilon_k\gamma g_2} \\
&= m_k - \gamma\epsilon_{k+1}\nabla_m F(m_k, \epsilon_k). \quad (35)
\end{aligned}
$$

**Mixture case**   We now extend NGD to a mixture of $N$ isotropic Gaussians with equal weights $1/N$:

$$p(x; \eta) = \frac{1}{N}\sum_{i=1}^{N}(2\pi)^{-d/2}\exp\left(\langle\eta^i, S(x)\rangle - A(\eta^i)\right)$$

with $\eta^i = \begin{pmatrix} \eta_1^i & \eta_2^i \end{pmatrix}^\top = \begin{pmatrix} \frac{m^i}{\epsilon^i} & -\frac{1}{2\,\epsilon^i} \end{pmatrix}^\top$ for $i = 1, \ldots, N$, which is a convex combination of exponential-family components, we apply NGD to each component.

Writing natural gradients in $(m^j, \epsilon^j)$-space gives

$$\begin{pmatrix} \frac{m_{k+1}^j}{\epsilon_{k+1}^j} \\ -\frac{1}{2\,\epsilon_{k+1}^j} \end{pmatrix} = \begin{pmatrix} \frac{m_k^j}{\epsilon_k^j} \\ -\frac{1}{2\,\epsilon_k^j} \end{pmatrix} - N\gamma\begin{pmatrix} g_1^j \\ g_2^j \end{pmatrix},$$

where for each $j$

$$\nabla_{\beta^j}\mathbf{f}([\beta^i]_{i=1}^N) = \begin{pmatrix} g_1^j \\ g_2^j \end{pmatrix}$$

and $\nabla_{m^j} F = g_1^j + 2\,m^j\,g_2^j, \nabla_{\epsilon^j} F = d\,g_2^j$. Equivalently, the updates are

$$\frac{1}{\epsilon_{k+1}^j} = \frac{1}{\epsilon_k^j} + \frac{2N\gamma}{d}\nabla_{\epsilon^j} F,$$

$$m_{k+1}^j = m_k^j - N\,\gamma\,\epsilon_{k+1}^j\,\nabla_{m^j} F.$$

*Remark* C.1.  Note that while $\mathrm{KL}(\cdot|\cdot)$ is known to be a Bregman divergence on the space of probability distributions over $\mathbb{R}^d$ [Aubin-Frankowski et al., 2022], it is not a Bregman divergence on $\mathbb{R}^d \times \mathbb{R}^{+*}$. Indeed, note that Eq (33) does not decouple the mean and variance terms, resulting in coupled updates in Eq (34)-(35).

# D   Approximation error for mixtures of isotropic Gaussians

In this section, we investigate the approximation error that can be achieved within the variational family we consider in this work. Previously, [Huix et al., 2024, Theorem 7] established that the approximation error of VI within the family of mixtures of Gaussian distributions with equal weights and constant isotropic covariance in the (reverse) Kullback-Leibler tends to $0$ as $N$ tends to infinity, under the assumption that the target distribution writes as an infinite mixture of these isotropic Gaussian components with same covariance. Below, we derive a similar result for our richer family, i.e., mixtures of (isotropic) Gaussian distributions with equal weights (and possibly different covariances). Note that we are deriving the results for mixtures of isotropic Gaussians, but the result and computations would the same for Gaussians with full covariance matrix.

**Assumption D.1.** There exists $p^*$ on $\mathbb{R}^d \times \mathbb{R}^{+*}$ such that the target $\pi$ writes as:

$$\pi := \int_\Theta k_\epsilon^m dp^\star(m, \epsilon), \tag{36}$$

where $k_\epsilon^m(x) := k_\epsilon(x - m)$ for any $x \in \mathbb{R}^d$.

Recall that we use the notation $\rho_N = k_\epsilon \otimes \mu = \int k_\epsilon^m d\hat{p}(m, \epsilon)$ with $\mu = \frac{1}{N}\sum_{i=1}^N \delta_{m^i}$ for an isotropic Gaussian mixture with $N$ components (where $\hat{p} = \frac{1}{N}\sum_{j=1}^N \delta_{(m^j, \epsilon^j)}$, and in the notation $k_\epsilon \otimes \mu$, $\epsilon$ is identified to the vector $(\epsilon_1, \ldots, \epsilon_N)$). We now state and prove our generalization of [Huix et al., 2024, Theorem 7].

**Theorem D.2.** *Let $\mathcal{C}^N = \left\{ \frac{1}{N}\sum_{j=1}^N \mathcal{N}(m^j, \epsilon^j \mathrm{I}_d), \ [m^j, \epsilon^j]_{j=1}^N \in (\mathbb{R}^d \times \mathbb{R}^{+*})^N \right\}$. Suppose that Assumption D.1 holds, then*

$$\min_{\rho \in \mathcal{C}^N} \mathrm{KL}(\rho|\pi) \leq C_\pi^2 \frac{\log(N) + 1}{N}, \quad where \quad C_\pi^2 = \int \frac{\int k_\epsilon^m(x)^2 dp^\star(m, \epsilon)}{\int k_\epsilon^m(x) dp^\star(m, \epsilon)} dx.$$

*Proof.* We denote

$$D_N = \min_{\rho \in \mathcal{C}^N} \mathrm{KL}(\rho|\pi), \quad \rho_N = \operatorname*{argmin}_{\rho \in \mathcal{C}^N} \mathrm{KL}(\rho|\pi).$$

For any $m \in \mathbb{R}^d$, we consider $\rho_{N+1}^{m,\epsilon} \in \mathcal{C}_{N+1}$ defined as

$$\rho_{N+1}^{m,\epsilon} = (1 - \alpha)\rho_N + \alpha k_\epsilon^m,$$

with $\alpha = \frac{1}{N+1}$. By definition of $D_N$, we have that, $D_{N+1} \leq \mathrm{KL}(\rho_{N+1}^{m,\epsilon}|\pi)$. Denoting $f(x) = x \log x$, we have $\mathrm{KL}(\rho_{N+1}^{m,\epsilon}|\pi) = \int f(r_{N+1}) d\pi$, where we define:

$$r_{N+1} := \frac{\rho_{N+1}^{m,\epsilon}}{\pi} = (1 - \alpha)\frac{\rho_N}{\pi} + \alpha \frac{k_\epsilon^m}{\pi} := r_0 + \alpha \frac{k_\epsilon^m}{\pi}.$$

Defining $B(x) = \frac{x \log x - x + 1}{(x-1)^2}$ for $x \in \mathbb{R}_+^* \backslash \{1\}$. By Lemma D.3, this function is decreasing; and since $r_{N+1}(x) \geq r_0(x) \ \forall x$, we have $B(r_{N+1}(x)) \leq B(r_0(x))$. It follows that

$$\begin{aligned}
f(r_{N+1}) = r_{N+1} \log(r_{N+1}) &\leq r_{N+1} - 1 + B(r_0)(r_{N+1} - 1)^2 \\
&= r_0 + \alpha \frac{k_\epsilon^m}{\pi} - 1 + B(r_0)(r_0 + \alpha \frac{k_\epsilon^m}{\pi} - 1)^2 \\
&= \alpha \frac{k_\epsilon^m}{\pi} + r_0 \log(r_0) + \alpha^2 \left(\frac{k_\epsilon^m}{\pi}\right)^2 B(r_0) + 2\alpha \frac{k_\epsilon^m}{\pi} B(r_0)(r_0 - 1). \tag{37}
\end{aligned}$$

Moreover, we have:

$$D_{N+1} = \int D_{N+1} dp^\star(m, \epsilon)$$

$$\leq \int \mathrm{KL}(\rho_{N+1}^{m,\epsilon} | \pi) dp^\star(m, \epsilon)$$

$$= \int \int f(r_{N+1}) d\pi dp^\star(m, \epsilon)$$

$$\leq \int \int \alpha \frac{k_\epsilon^m}{\pi} d\pi dp^\star(m, \epsilon) + \int \int r_0 \log(r_0) d\pi dp^\star(m, \epsilon)$$

$$+ \int \int \alpha^2 \left( \frac{k_\epsilon^m}{\pi} \right)^2 B(r_0) d\pi dp^\star(m, \epsilon)$$

$$+ \int \int 2\alpha \frac{k_\epsilon^m}{\pi} B(r_0)(r_0 - 1) d\pi dp^\star(m, \epsilon) \quad \text{from (37)}$$

$$= \alpha + \int r_0(x) \log(r_0(x)) d\pi(x) + \alpha^2 \int \int \frac{k_\epsilon^m(x)^2}{\pi(x)} B(r_0(x)) dx \, dp^\star(m, \epsilon)$$

$$+ 2\alpha \int \int k_\epsilon^m B(r_0(x))(r_0(x) - 1) dx \, dp^\star(m, \epsilon)$$

$$= \alpha + \int r_0(x) \log(r_0(x)) d\pi(x) \tag{a}$$

$$+ \alpha^2 \int \int \frac{k_\epsilon^m(x)^2}{\pi(x)} B(r_0(x)) dx \, dp^\star(m, \epsilon) \tag{b}$$

$$+ 2\alpha \int B(r_0(x))(r_0(x) - 1)\pi(x) dx. \tag{c}$$

We first observe that we can write (a) in function of $D_N$. Indeed, $r_0(x) = (1 - \alpha)\frac{\rho_N}{\pi}$, so

$$\text{(a)} = \int r_0(x) \log(r_0(x)) d\pi(x)$$

$$= \int (1 - \alpha)\frac{\rho_N}{\pi} \log\left( (1 - \alpha)\frac{\rho_N}{\pi} \right) d\pi$$

$$= (1 - \alpha) \int \rho_N(x) \log\left( \frac{\rho_N(x)}{\pi(x)} \right) dx + (1 - \alpha) \log(1 - \alpha)$$

$$= (1 - \alpha) D_N + (1 - \alpha) \log(1 - \alpha).$$

For the second term (b), we have that $\lim_{x \to 0^+} B(x) = 1$ and since $B$ decrease, $B(x) \leq 1$, thus $B(r_0(x)) \leq 1$, this implies :

$$\text{(b)} = \alpha^2 \int \int \frac{k_\epsilon^m(x)^2}{\pi(x)} B(r_0(x)) dx \, dp^\star(m, \epsilon) \leq \alpha^2 \int \int \frac{k_\epsilon^m(x)^2}{\pi(x)} dx \, dp^\star(m, \epsilon)$$

$$= \alpha^2 C_\pi^2.$$

And for the third term (c), we have that $B(x)(x - 1) \leq \sqrt{x} - 1$, see Lemma D.4. Thus,

$$\text{(c)} = 2\alpha \int B(r_0(x))(r_0(x) - 1)\pi(x) dx \leq 2\alpha \int \left( \sqrt{r_0(x)} - 1 \right) \pi(x) dx$$

$$= 2\alpha \int \sqrt{(1 - \alpha)\frac{\rho_N(x)}{\pi(x)}} \pi(x) dx - 2\alpha$$

$$= 2\alpha\sqrt{1 - \alpha} \int \sqrt{\rho_N(x)\, \pi(x)} dx - 2\alpha$$

$$= 2\alpha\sqrt{1 - \alpha} \left( 1 - H^2(\rho_N, \pi) \right) - 2\alpha$$

$$\leq 2\alpha\sqrt{1 - \alpha} - 2\alpha,$$

where we have used the definition of the squared Hellinger distance $H^2(f, g) = 1 - \int \sqrt{f(x)g(x)}dx$ and the property stating that for any densities of probability $f, g, 0 \leq H^2(f, g) \leq 1$.

Finally, we have

$$D_{N+1} \leq \alpha + (1 - \alpha)D_N + (1 - \alpha)\log(1 - \alpha) + \alpha^2 C_\pi^2 + 2\alpha\left(\sqrt{1 - \alpha} - 1\right)$$
$$\leq (1 - \alpha)D_N + \alpha^2 C_\pi^2, \tag{38}$$

using Lemma D.5, stating that $\alpha + (1 - \alpha)\log(1 - \alpha) + 2\alpha\left(\sqrt{1 - \alpha} - 1\right) \leq 0$.

The previous inequality (38) is true for any $n \geq 0$ and recalling that $\alpha = \frac{1}{n+1}$ we have,

$$D_{n+1} \leq (1 - \alpha)D_n + \alpha^2 C_\pi^2$$
$$(n + 1)D_{n+1} - nD_n \leq \frac{1}{n+1}C_\pi^2$$
$$\sum_{n=0}^{N-1}(n + 1)D_{n+1} - nD_n \leq C_\pi^2 \sum_{n=0}^{N-1}\frac{1}{n+1}$$
$$ND_N \leq C_\pi^2(\log(N) + 1)$$
$$D_N \leq C_\pi^2 \frac{\log(N) + 1}{N},$$

where the Harmonic number $\sum_{n=0}^{N-1}\frac{1}{n+1}$ has been bounded by $\log(N) + 1$. $\qquad\square$

In the proof above we used the following lemmas from [Huix et al., 2024] (Lemma 8 to 10 therein). We provide their proofs for completeness.

**Lemma D.3.** *The function* $B(x) = \frac{x \log x - x + 1}{(x-1)^2}$ $\quad \forall x \in \mathbb{R}^{+*}\backslash\{1\}$, *is decreasing.*

*Proof.* For all $x \in \mathbb{R}^{+*}\backslash\{1\}$ the gradient of $B$ writes:

$$\nabla B(x) = \frac{(x - 1)\log x - 2(x \log x - x + 1)}{(x - 1)^3}$$

- For $x \in (0, 1)$, the denominator is strictly negative and the numerator strictly positive, thus $\nabla B(x) \leq 0$.

- For $x \in (1, \infty)$, the denominator is stricly positive and the numerator is strictly negative, thus $\nabla B(x) \leq 0$.

So $B$ is decreasing on both intervals, and $\lim_{x \to 1^-} = \frac{1}{2}$ and $\lim_{x \to 1^+} = \frac{1}{2}$ by Hospital's rule. $\quad\square$

**Lemma D.4.** *The function $B$ satisfies:* $B(x)(x - 1) \leq \sqrt{x} - 1$ $\quad \forall x \in \mathbb{R}^{+*}\backslash\{1\}$.

*Proof.* Let $C(x) := B(x)(x - 1) = \frac{x \log x - x + 1}{x - 1}$

- For $x \in (0, 1)$, $\log x \geq \frac{x-1}{\sqrt{x}}$ implies $\frac{\log x}{x-1} \leq \frac{1}{\sqrt{x}} \Rightarrow \frac{x \log x}{x-1} \leq \frac{x}{\sqrt{x}} = \sqrt{x}$,

- For $x \in (1, \infty)$, $\log x \leq \frac{x-1}{\sqrt{x}}$ implies $\frac{\log x}{x-1} \leq \frac{1}{\sqrt{x}} \Rightarrow \frac{x \log x}{x-1} \leq \frac{x}{\sqrt{x}} = \sqrt{x}$.

and $C(x) - \sqrt{x} - 1 = \frac{x \log x}{x-1} - \sqrt{x} \leq 0$. $\qquad\square$

**Lemma D.5.** *Let consider* $\alpha = \frac{1}{n+1}$ $\quad \forall n \in \mathbb{N}$, *then* $\alpha + (1-\alpha)\log(1 - \alpha) + 2\alpha\left(\sqrt{1 - \alpha} - 1\right) \leq 0$.

*Proof.* We have:

$$\alpha + (1-\alpha)\log(1-\alpha) + 2\alpha\left(\sqrt{1-\alpha}-1\right) = -\alpha + (1-\alpha)\log(1-\alpha) + 2\alpha\sqrt{1-\alpha}$$
$$\leq \alpha\left(\alpha - 2 + 2\sqrt{1-\alpha}\right)$$
$$\leq \alpha - 2 + 2\sqrt{1-\alpha}$$
$$\leq 0,$$

using $\log(1-\alpha) \leq -\alpha$ and the fact that $\alpha = 1/(n+1) \leq 1$ for the first and second inequality. For the last inequality we have used the fact that the before last expression is decreasing and is equal to $0$ when $\alpha$ goes to $0$. $\qquad\square$

# E  Additional experiments and details

## E.1  Updates for the full-covariance matrices scheme of Lambert et al. [2022, Section 5.2]

In this section we detail the updates for the full-covariance scheme of Lambert et al. [2022, Section 5.2]. The parameter space is $\Theta = \mathbb{R}^d \times \mathbb{S}^d_{++}$ (the space of means and covariance matrices). Consider initializing this evolution at a finitely supported distribution $p_0$:

$$p_0 = \frac{1}{N}\sum_{i=1}^{N}\delta_{\theta_0^{(i)}} = \frac{1}{N}\sum_{i=1}^{N}\delta_{(m_0^{(i)},\Sigma_0^{(i)})}$$

It has been checked in Lambert et al. [2022] that the system of ODEs thus initialized maintains a finite mixture distribution:

$$p_t = \frac{1}{N}\sum_{i=1}^{N}\delta_{\theta_t^{(i)}} = \frac{1}{N}\sum_{i=1}^{N}\delta_{(m_t^{(i)},\Sigma_t^{(i)})},$$

where the parameters $\theta_t^{(i)} = (m_t^{(i)}, \Sigma_t^{(i)})$ evolve according to the following interacting particle system, for $i \in [N]$

$$\dot{m}_t^{(i)} = -\mathbb{E}\nabla\ln\frac{\nu_t}{\pi}\left(Y_t^{(i)}\right),$$
$$\dot{\Sigma}_t^{(i)} = -\mathbb{E}\nabla^2\ln\frac{\nu_t}{\pi}\left(Y_t^{(i)}\right)\Sigma_t^{(i)} - \Sigma_t^{(i)}\mathbb{E}\nabla^2\ln\frac{\nu_t}{\pi}\left(Y_t^{(i)}\right),$$

where $Y_t^{(i)} \sim p_{\theta_t^{(i)}}$ and $\nu_t = \int \mathcal{N}_\theta dp_t(\theta)$. In their experiments, the ODEs were solved using a fourth-order Runge–Kutta scheme. In our experiments we applied the BW SGD described in Lambert et al. [2022, Algorithm 1] .

## E.2  Experimental details

As mentioned in Section 6, we detail here our experimental setup and hyperparameters whose values are provided in Table 1.

**Initialization of the variational mixture:** For $N \in \mathbb{N}^*$ a given number of components, we initialize the variational mixture by sampling the means in a ball of size $[-s, s]^d$, where $s \in \mathbb{R}^{+*}$, and setting each covariance matrix to $r\,I_d$, where $r \in \mathbb{R}^{+*}$. For the GD algorithm (mean optimization only, following Huix et al. [2024]), the variances are initialized in the same way but kept fixed during optimization.

**Optimization hyperparameters:** We set the step-size $\gamma$, the number of iteration $n_{\text{iter}}$, the number of Monte Carlo samples to $B_{\text{grad}} = 10$ for gradient estimation, and to $B_{\text{KL}} = 1000$ for the KL objective estimation.

**Normalizing Flows:** For the NF baseline, we used a simplified RealNVP architecture [Dinh et al., 2017], based on the code available at `https://github.com/marylou-gabrie/tutorial-sampling-enhanced-w-generative-models`, with $b = 2$ coupling layers and hidden

Table 1: Hyperparameters

| Exp. | MOG Init. $s$ | $r$ | Optim $\gamma$ | $n_{\text{iter}}$ | Targets |
|---|---|---|---|---|---|
| **Figure 1** | | | | | |
| `mog` | 15 | 2 | $10^{-1}$ | $10^3$ | $\{s_{\text{tg}}, r_{\text{tg}}, N_{\text{tg}}\} = \{8, 5, 5\}$ |
| **Figure 3** | | | | | |
| `mog` | $10^2 d^{-1}$ | 10 | $10^{-2} d^{-1}$ | $10^3$ | $\{s_{\text{tg}}, r_{\text{tg}}, N_{\text{tg}}\} = \{10^2 d^{-1}, 5, 5\}$ |
| **Figure 2** | | | | | |
| `mog` | 30 | 100 | $10^{-2}$ | $10^4$ | $\{s_{\text{tg}}, r_{\text{tg}}, N_{\text{tg}}\} = \{10, 1, 10\}$ |
| **Figure 4** | | | | | |
| `breast_cancer` | 20 | 10 | $10^{-2}$ | $10^4$ | $\sigma^2_{prior} = 100$ |
| `wine` | 20 | 10 | $10^{-2}$ | $10^2$ | $\sigma^2_{prior} = 100$ |
| `boston` | 10 | 10 | $10^{-6}$ | $10^4$ | $\{\sigma^2_{prior}, h\} = \{10, 50\}$ |
| **Figure 6** | | | | | |
| `funnel` (a) | 5 | 0.5 | $10^{-2}$ | $10^4$ | $\sigma^2 = 1.2$ |
| `sinh-arcsinh` (b-1) | 10 | 2 | $10^{-3}$ | $10^4$ | skw $= (-0.2, -0.2)$ |
| `sinh-arcsinh` (b-2) | 10 | 2 | $10^{-3}$ | $10^4$ | skw $= (-0.2, -0.5)$ |
| `mog` (c-1) | 10 | 5 | $10^{-2}$ | $10^4$ | $\{\text{pt}, r_{\text{tg}}, N_{\text{tg}}\} = \{3, 2, 4\}$ |
| `mog` (c-2) | 10 | 5 | $10^{-2}$ | $10^4$ | $\{\text{pt}, r_{\text{tg}}, N_{\text{tg}}\} = \{4, 1, 4\}$ |
| `mog` (c-3) | 10 | 5 | $10^{-2}$ | $10^4$ | $\{\text{pt}, r_{\text{tg}}, N_{\text{tg}}\} = \{3, 2, 4\}$ |
| **Figure 10** | | | | | |
| `mog` | 30 | 100 | $10^{-3}$ | $10^4$ | $\{s_{\text{tg}}, r_{\text{tg}}, N_{\text{tg}}\} = \{20, 5, 5\}$ |
| **Figure 11** | | | | | |
| `mog` | 30 | 100 | $10^{-3}$ | $10^4$ | $\{s_{\text{tg}}, r_{\text{tg}}, N_{\text{tg}}\} = \{10, 10, 10\}$ |

dimension $h = 124$, which yields a Neural Network (NN) with 4976 parameters in dimension $d = 2$. Our isotropic mixture model has $N(d + 1) = 3N$ parameters, even for large $N$, the NF model remains more complex and costly to optimize. Therefore, for each target distribution, we tuned the learning rate and number of iterations for the NF method separately, rather than using the same settings as for the VI mixture methods, since their optimization dynamics differ significantly.

**MOG targets:** To generate target MoG distributions, as in the initalization of variational MOG, we fix $s_{\text{tg}}$ and $N_{\text{tg}}$. Each component covariance matrix is constructed by sampling a random symmetric positive-definite matrix (full diagonal or isotropic) and scaling it by $r_{\text{tg}}$. We draw raw weights uniformly in $\{1, \ldots, 2N_{\text{tg}}\}$ and normalize them to one. In Figure 6(c) all weights are equal except in case (c-3), where one component has weight 0.1 and the remaining ones share the remaining mass equally and the component means are placed at $(\pm \text{pt}, \pm \text{pt})$.

**Datasets:** We have used popular datasets from the UCI repository, as well as MNIST. The training ratio has been set to 0.5 for UCI datasets and 0.8 for MNIST.

**Computational resources:** All experiments (except MNIST) were conducted on a MacBook Air (M3, 2024) with an Apple M3 processor and 16 GB of RAM. The MNIST experiments were run on an NVIDIA 50-90 GPU. Experiment runtimes ranged from a few seconds to up to two hours.

## E.3 Additional 2-D examples

We present more experiments on 2D synthetic target distributions on Figure 6. These target distributions are defined below.

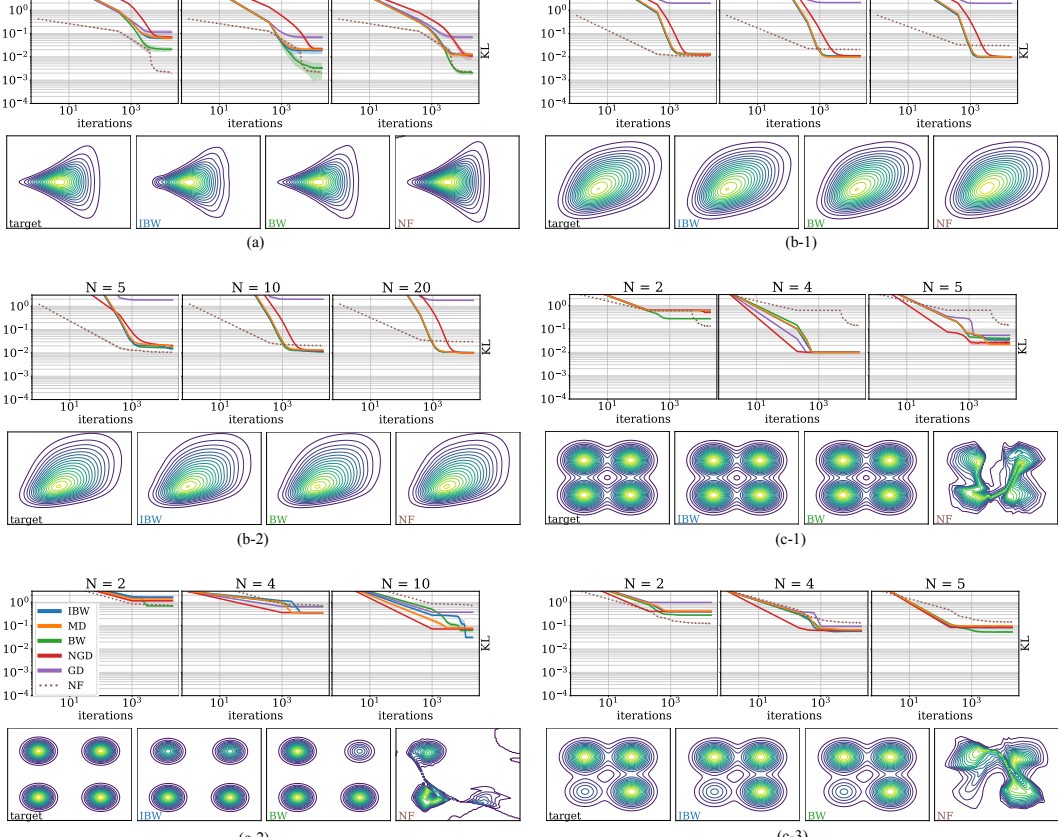

Figure 6: Evolution of the KL divergence over iterations together with the optimized variational mixture density on different type of target distribution: (a) Funnel, (b) sinh-arcsinh normal distribution and (c) Gaussian mixture. Optimization performed with different methods (IBW, MD, BW, NGD, GD and NF) and varying $N$ values.

**Funnel distribution:** The funnel distribution [Neal, 2003] in dimension $d = 2$ has density

$$p(x_1, x_2) = \mathcal{N}(x_1; 0, \sigma^2) \ \times \ \mathcal{N}(x_2; 0, e^{x_1}),$$

for $x = (x_1, x_2) \in \mathbb{R}^2$. We follow the setting of Cai et al. [2024a] by fixing $\sigma^2 = 1.2$. Although unimodal, this "funnel" shape is difficult to capture with isotropic Gaussians. We experimented with $N = 5, 20, 40$ components, but even for large $N$, our isotropic mixtures struggled, and the BW and NF methods still outperformed them.

**Sinh-arcsinh normal distribution:** This distribution [Pewsey, 2009] applies a sinh–arcsinh transformation to a multivariate Gaussian to control the skewness skw and tail weight $\tau$. Let

$$Z_0 \sim \mathcal{N}(m, \Sigma), \qquad Z = \sinh(\tau \sinh^{-1}(Z_0) - \text{skw}).$$

In our experiments, we use $\tau = (0.8, 0.8)$, $m = (0, 0)$, $\Sigma = \begin{pmatrix} 1 & 0.4 \\ 0.4 & 1 \end{pmatrix}$, and vary the skew parameter skw as specified in Table 1.

In Figure 7, we visualize the optimized Gaussian components of the target density, highlighting the advantages of allowing each component to have its own $\epsilon$ value.

### E.4 Bias induced by uniform weights mixture

We investigate the bias induced by our choice of the variational family, i.e. mixtures (of isotropic Gaussians) with uniform weights. Our goal in this section is to analyze how such a constraint affects the approximation of a bimodal target distribution with highly unbalanced mode weights.

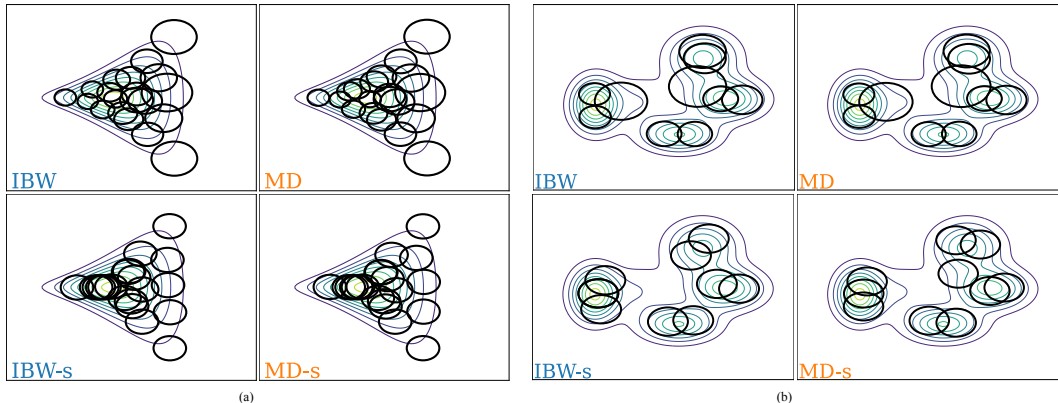

Figure 7: Optimized isotropic Gaussians (represented as circles) over the target distributions for the proposed methods and their shared-variance variants. (a) corresponds to the Funnel distribution in Figure 6(a); (b) corresponds to the target used in Figure 1

We consider as the target a two-dimensional Gaussian mixture with two components and weights $(p, 1 - p)$ where $p = 0.1$. We seek the optimal approximation within our variational family of $N$ isotropic Gaussian components with uniform weights. To mitigate optimization issues and being trapped in a local minima, we initialize the means of our variational approximation by sampling the initial components means in two small balls centered around each target mode (where the number of initial means sampled in a ball is proportional to $Np$ and $N(1 - p)$ respectively).

As expected, when $N = 2$, the weights constraint induces a strong bias: the approximation either ignores the low-weight mode or assigns equal mass to both modes. However, as $N$ increases, this bias progressively vanishes. For sufficiently large $N$, the model reallocates more components to the dominant mode and fewer to the lighter one, effectively recovering the correct proportions despite the uniform-weight constraint.

To quantify how well the variational mixture captures both the mode locations and their effective weights, we drew $B = 10,000$ samples from both the variational distribution and the target distribution and applied K-means clustering with $n_{\text{clusters}} = 2$, then analyzed the cluster assignments and proportions. The reported errors are: $E_1$ the mean Euclidean distance between the target and variational modes, and $E_2$ the absolute difference between their corresponding cluster weights. The first metric captures whether the variational modes are well located, while the second one measures whether their relative proportions are well captured. Each experiment is repeated 10 times; we report the average result and error bars. Figure 8 shows that both errors rapidly decrease with $N$, demonstrating that uniform-weight Gaussian mixtures can approximate highly unbalanced multimodal targets remarkably well when sufficiently overparameterized. We also display the estimated variational density and Gaussians components (circles) along with the target density.

### E.5 Additional high-dimensional mixtures

We first provide the full marginals of the experiment described in Figure 2 in Figure 9. We also performed experiments in $d = 10$ Figure 10 and $d = 50$ Figure 11.

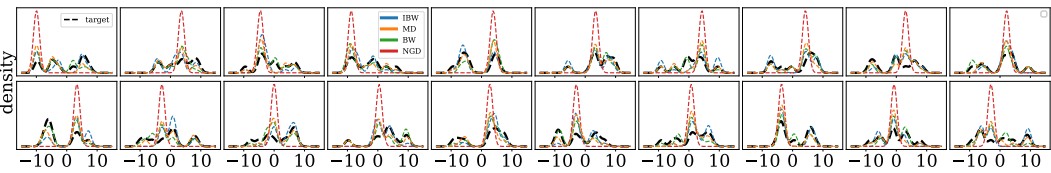

Figure 9: Marginals for MD, IBW, BW and NGD ($d = 20$).

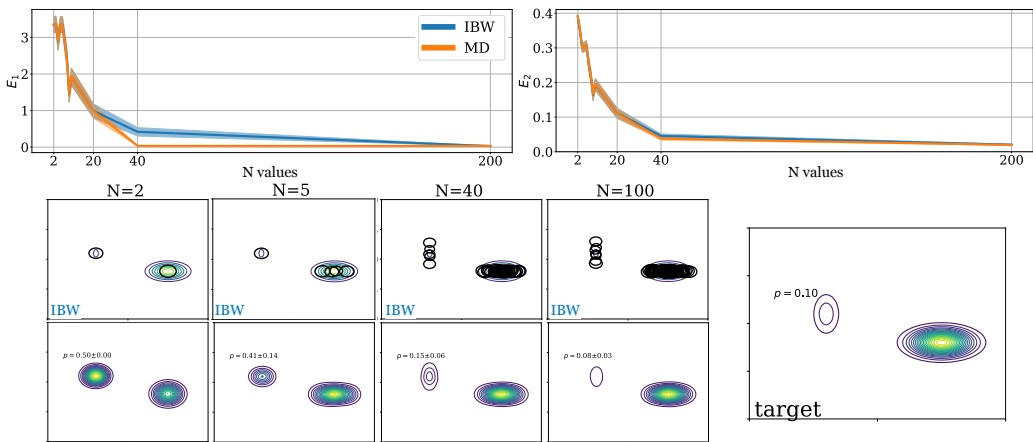

Figure 8: Errors, visualization and estimation of the bias induced by the uniform weights constraint as a function of $N$ for a two dimensional unbalanced mixture of Gaussians target.

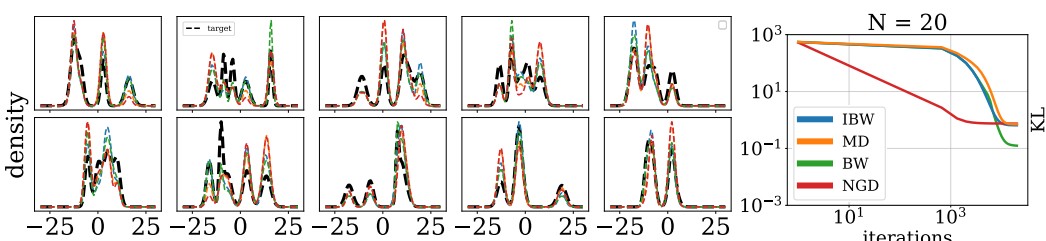

Figure 10: Marginals (left) and KL objective (right) for MD, IBW, BW and NGD (d = 10).

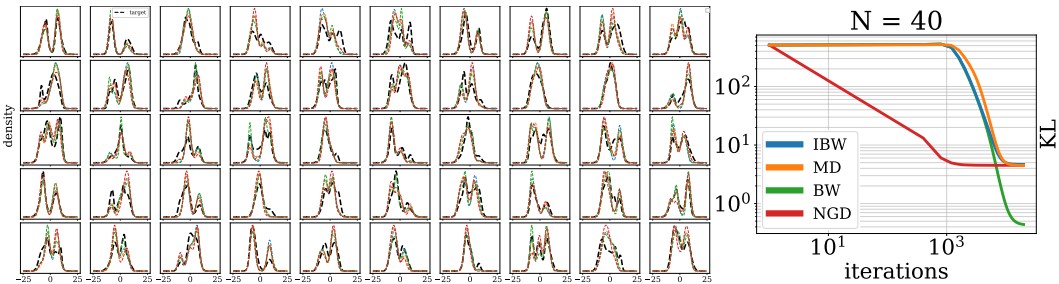

Figure 11: Marginals (left) and KL objective (right) for MD, IBW, BW and NGD (d = 50).

## E.6   More details on the Bayesian inference examples

In this section we provide some background on the Bayesian inference examples of Section 6 as well as additional experiments.

### E.6.1   Definition of the target distributions

Let $\mathcal{D} = \{(x_i, y_i)\}_{i=1}^n$ be a labeled dataset, where $x_i \in \mathbb{R}^d$ and $y_i$ is the associated label.

**Binary logistic regression:**   We model the probability of a binary label $y_i \in \{0, 1\}$ given $x_i$ and parameter $z \in \mathbb{R}^d$ by

$$\pi(y_i \mid x_i, z) = \sigma(x_i^\top z)^{y_i} (1 - \sigma(x_i^\top z))^{1-y_i},$$

where $\sigma(t) = 1/(1 + e^{-t})$ is the logistic function. The likelihood is

$$\mathcal{L}(\mathcal{D} \mid z) = \prod_{i=1}^n \pi(y_i \mid x_i, z),$$

and the log-likelihood is

$$\ell(\mathcal{D} \mid z) = \sum_{i=1}^{n} \log \pi(y_i \mid x_i, z) = \sum_{i=1}^{n} \left[ y_i \left( x_i^\top z \right) - \log \left( 1 + e^{x_i^\top z} \right) \right].$$

With a Gaussian prior $\pi(z) = \mathcal{N}(0, \sigma_{prior}^2 I_d)$, the posterior is

$$\pi(z \mid \mathcal{D}) \; \propto \; \mathcal{L}(\mathcal{D} \mid z) \, \pi(z) \tag{39}$$

and its gradient is

$$\nabla_z \log \pi(z \mid \mathcal{D}) = \sum_{i=1}^{n} \left( y_i - \sigma(x_i^\top z) \right) x_i \; - \; \frac{z}{\epsilon_z}.$$

**Multi class logistic regression:** For $L$ classes, let $z = (z_1, \ldots, z_L)$ with each $z_l \in \mathbb{R}^d$. Then

$$\pi(y_i = l \mid x_i, z) = \frac{\exp\left( x_i^\top z_l \right)}{\sum_{l=1}^{L} \exp\left( x_i^\top z_l \right)}, \qquad l = 1, \ldots, L.$$

**Linear regression:** The classical linear model is

$$y_i = z^\top x_i + \xi_i, \quad \xi_i \sim \mathcal{N}(0, \sigma^2),$$

so that

$$y_i \sim \mathcal{N}\left( x_i^\top z, \, \sigma^2 \right).$$

and the ordinary least squares estimator is

$$\hat{z} \; = \; \arg\min_{z \in \mathbb{R}^d} \sum_{i=1}^{n} (y_i - z^\top x_i)^2.$$

for which we are able to find a close form when $X = (x_i)_{i=1}^{n}$ is invertible.

In our Bayesian setting we aim at finding a distribution on $z$, put a prior on it, and approximate the resulting posterior $\pi(z \mid \mathcal{D})$ via variational inference.

**Bayesian Neural Network:** In the Bayesian neural network (BNN) setting, the linear predictor $z^\top x_i$ is replaced with a neural network output $f(x_i \mid z)$ and model

$$y_i \sim \mathcal{N}\left( f(x_i \mid z), \, \sigma^2 \right).$$

In our experiments we use a single hidden layer with $h$ hidden units, ReLU activation function and output dimension $c$. Thus, the dimension of parameters and thus of the problem is

$$d = h \left( d_{\text{data}} + 1 \right) \; + \; c \left( h + 1 \right).$$

For a $L$-class classification task, $c = L$ and the BNN output class probabilities $\pi(y_i = l \mid x_i, z) = f(x_i \mid z)_l$.

Once the variational approximation to the posterior is optimized, we can make predictions by Bayesian model averaging:

$$p(y \mid x) = \int \pi(y \mid x, z) \, \pi_{\text{post}}(z) \, dz, \quad \text{or} \quad \hat{y} = \int f(x \mid z) \, \pi_{\text{post}}(z) \, dz.$$

When $d$ is large (e.g. MNIST, where $d \approx 10^5$), sampling or expectation under a full $d$-dimensional mixture becomes too expensive. To address this, we adopt a mean-field-style approximation: we model the posterior as a product of identical univariate Gaussian-mixture marginals,

$$z_j \; \sim \; \frac{1}{N} \sum_{i=1}^{N} \mathcal{N}\left( m^i[j], \epsilon^i \right), \qquad j = 1, \ldots, d,$$

so that all $d$ dimensions share the same $N$-component mixture. This reduces both memory and computational cost while retaining multimodality in each coordinate. We updated $m^i, \epsilon^i$ using the

presented algorithms. In this setting, we follow a classical deep-learning framework. We use a single-layer neural network with $h = 256$ hidden units and ReLU activation. The means of the variational mixture of Gaussians are initialized by sampling from a normal distribution, and a Gaussian prior is placed on the model parameters.

**Laplace Approximation:** In the Bayesian setting, a well-known approach to approximate the posterior distribution is to fit it with a Gaussian. Given the posterior $\pi(z|\mathcal{D})$ defined in Equation (39), Laplace approximation methods consider a Gaussian $\mathcal{N}(z^\star, \Sigma^\star)$ where

$$z^\star = \operatorname*{argmin}_{z \in \mathbb{R}^d} U(z), \quad U(z) := -\log \pi(z|\mathcal{D}) = -\ell(\mathcal{D}|z) - \log \pi(z),$$

is the Maximum a Posteriori (MAP), and $\Sigma^\star$ is the inverse of the hessian of $U$ evaluated at $z^\star$. Indeed, performing a second-order Taylor expansion of U around $z^\star$ yields:

$$U(z) \approx U(z^\star) + (z - z^\star)^T \underbrace{\nabla U(z^\star)}_{=0} + \frac{1}{2}(z - z^\star)^T H(z - z^\star), \quad H = \nabla^2 U(z^\star)$$

which corresponds to the negative log of a gaussian with mean $z^\star$ and covariance $\Sigma^\star := H^{-1}$. The MAP estimate can be obtained with standard Euclidean optimization methods (e.g. gradient descent). However, in high-dimensional settings, the Hessian $H$ is typically too large to compute or invert directly, so it must be approximated. In our experiments, we focus on two practical approximations: the Diagonal approximation and the Kronecker-Factored (K-FAC) approximation [Ritter et al., 2018].

- Diagonal approximation:

$$H = \operatorname{diag}(h_1, \ldots, h_d), \quad h_i = \left. \frac{\partial^2 U(z)}{\partial z_i^2} \right|_{z=z^\star}$$

- K-FAC approximation:

$$H \approx \operatorname{diag}(H_1, \ldots, H_L), \quad H_\ell \approx G_\ell \otimes A_\ell,$$

$$A_\ell := \mathbb{E}_{\mathcal{D}}\left[ a_{\ell-1} a_{\ell-1}^\top \right], \quad G_\ell := \mathbb{E}_{\mathcal{D}}\left[ g_\ell g_\ell^\top \right],$$

where $a_{\ell-1}$ are layer inputs and $g_\ell$ the backpropagated pre-activation gradients; expectations are empirical over $\mathcal{D}$ and all quantities are evaluated at $z^\star = \arg\min_z U(z)$.

### E.6.2   Real data experiments

**Comparison with Laplace Approximation.**   We compare our method using $N = 1$ and $N = 5$ Gaussian components with Laplace approximations on MNIST. On the training set, all methods achieve similar accuracy and negative log-likelihood (NLL), while on the test set our approach yields much better performance, indicating it generalizes better with multiple modes. To illustrate the effect of using multiple Gaussian components, we perform PCA on the means obtained with MD and IBW for $N = 1$ and $N = 5$, along with the MAP estimate used in the Laplace approximation. As shown in Figure 13, the means for $N = 5$ do not collapse, highlighting the richer posterior structure captured by multiple components.

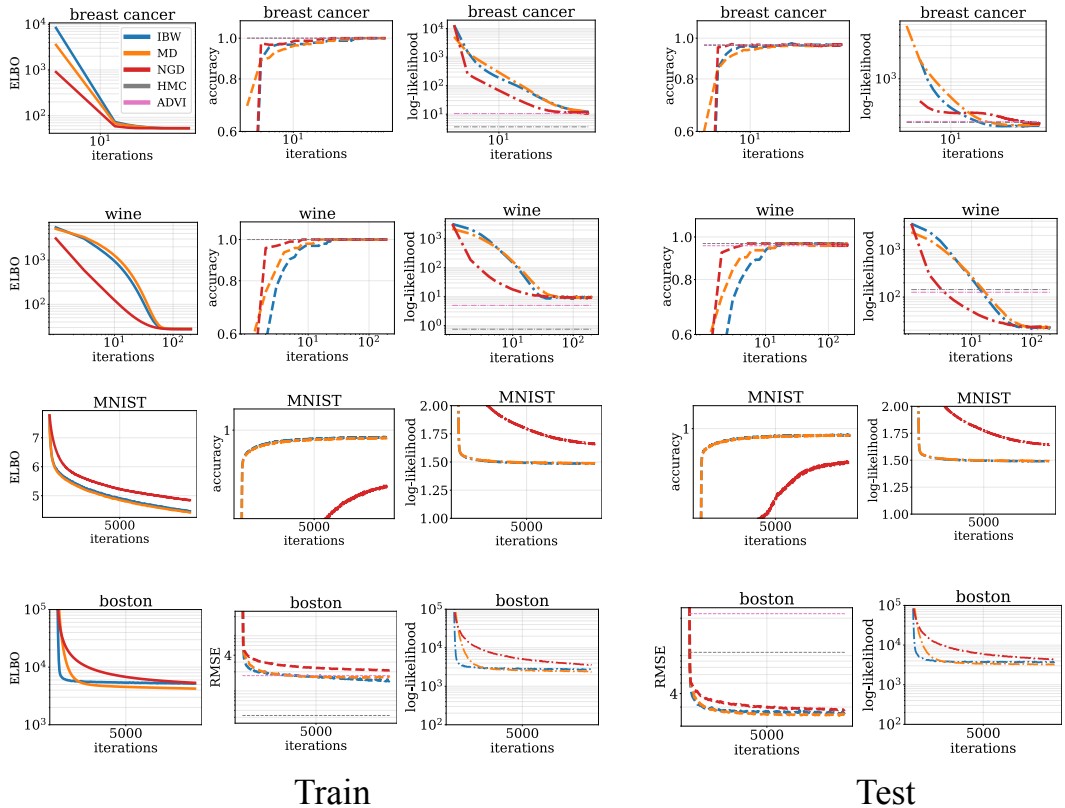

Figure 12: Evolution of the ELBO, accuracy (or RMSE), and log-likelihood over iterations on the **train** set (left) and **test** set (Tight) for the `breast_cancer` (upper row), `wine` (middle row), MNIST and `boston` (bottom row) datasets for $N = 5$.

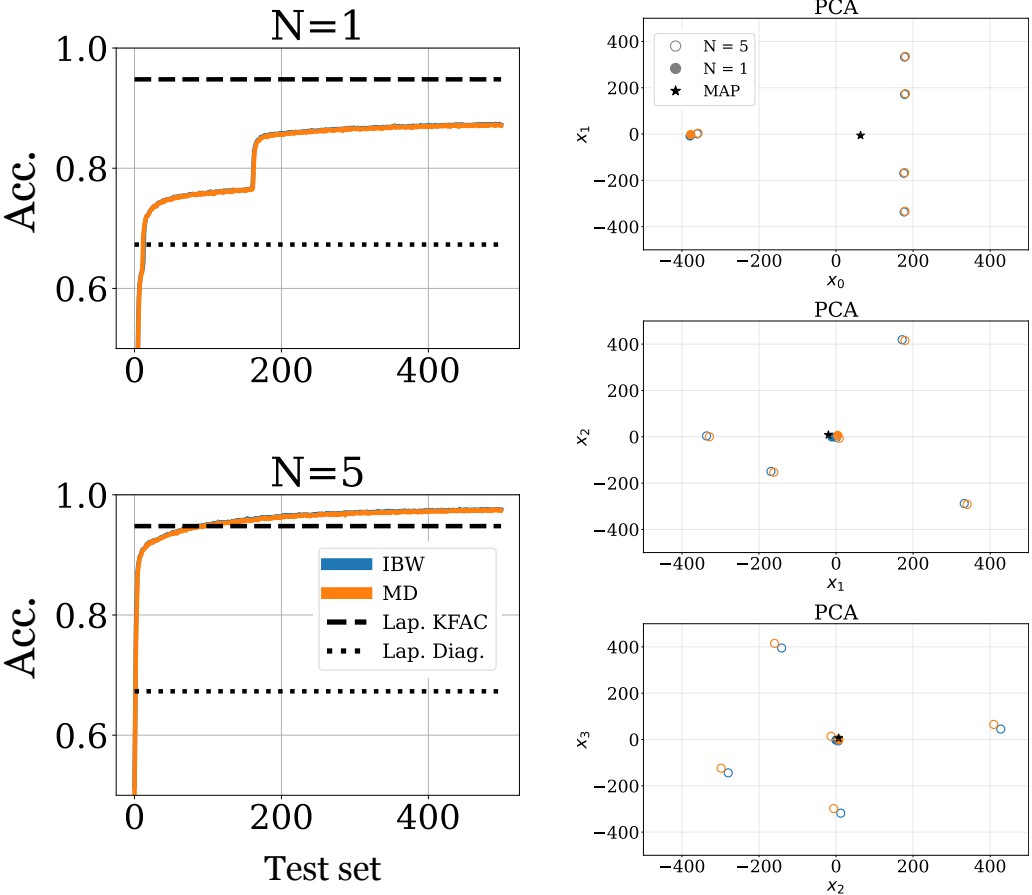

Figure 13: Comparison of Laplace and our methods IBW and MD on MNIST. Left: train and test accuracies for $N = 1$ and $N = 5$. Right: Principal Component Analysis (PCA) projections on the three principal axis of the Gaussian component means obtained with MD, IBW, and Laplace approximations.

