# OpenReview forum: "Variational Inference with  Mixtures of Isotropic Gaussians"
_NeurIPS.cc/2025/Conference — NeurIPS 2025 poster_

### Official Review · Reviewer_Fvva · 2025-06-19

**Clarity:** 3
**Significance:** 4
**Originality:** 4
**Rating:** 5
**Confidence:** 3

**Summary:**

The authors present a general variational framework based on mixtures of isotropic Gaussians. Despite being less flexible than the counterparts with full covariance matrices, the proposal is also capable of approximating multimodal posteriors, while being much more efficient in terms of computational and memory costs. In this context, the paper derives two variational inference algorithms, which are evaluated in common approximate inference tasks. The experimental comparison with other strategies from the literature show promising results.

**Questions:**

- At which scenarios the introduced inference algorithms (IBW and MD) are more indicated? Guidelines for choosing between the two schemes would be a welcomed inclusion to the work.

- Line 252: The decoupling of the updates on the means and variances is an advantage or a disadvantage here?

- Some references are incomplete or show the arXiv versions, instead of published versions, e.g., lines 408, 437, 442, 447.

**Ethical Concerns:**

["NO or VERY MINOR ethics concerns only"]

**Final Justification:**

The authors addressed my points and I maintain my favorable score.

**Limitations:**

The authors state the limitations of their work.

**Paper Formatting Concerns:**

None.

**Quality:**

4

**Strengths And Weaknesses:**

The paper is well organized and written. Although it demands a substantial background from the reader, the authors provide useful references across all the sections. The related work is up to date and highlight the introduced contributions. The experimental section covers useful tasks (low-mid dimension inference and approximate posterior for standard machine learning models). I also praise the authors for the provided code and comprehensive supplementary material.

I did not find great weaknesses in the paper, but I did miss a comparison with more recent approximate inference methods based on deep generative models. The authors do use a simple Normalizing Flow as a baseline, but strategies based on GFlowNets [1,2], Diffusion models [3,4], or Flow Matching [5] could also be considered or at least included in the discussion.

References

[1] Nikolay Malkin, Salem Lahlou, Tristan Deleu, Xu Ji, Edward J. Hu, Katie Everett, Dinghuai Zhang, Yoshua Bengio: GFlowNets and variational inference. ICLR 2023

[2] Salem Lahlou, Tristan Deleu, Pablo Lemos, Dinghuai Zhang, Alexandra Volokhova, Alex Hernández-García, Léna Néhale Ezzine, Yoshua Bengio, Nikolay Malkin: A theory of continuous generative flow networks. ICML 2023: 18269-18300

[3] Francisco Vargas, Will Sussman Grathwohl, Arnaud Doucet: Denoising Diffusion Samplers. ICLR 2023

[4] Wasu Top Piriyakulkij, Yingheng Wang, Volodymyr Kuleshov: Denoising Diffusion Variational Inference: Diffusion Models as Expressive Variational Posteriors. AAAI 2025: 19921-19930

[5] Jonas Wildberger, Maximilian Dax, Simon Buchholz, Stephen R. Green, Jakob H. Macke, Bernhard Schölkopf: Flow Matching for Scalable Simulation-Based Inference. NeurIPS 2023

---

> ### Author Rebuttal · Authors · 2025-07-31
>
> We thank the reviewer for his very positive comments about our paper. His questions and pointed weaknesses are commented below.
>
> **Weakness - about additional neural-network based baselines**
>
> We thank the reviewer for the references, most of them we were not familiar with. We originally included baselines that were relatively standard in recent VI papers, and our point when including NF was to highlight that the use of Neural Networks induced a significant computational overhead compared to our scheme, while yielding unsatisfying results. Yet we acknowledge that the alternative neural-network based approaches you mention could perform very differently and would be interesting to compare.  We will mention them in the revised version in the discussion.
>
> **Q1: choice of IBW vs MD**
>
> See answer to Q4 of Reviewer cDha.
>
> **Q2: about coupled updates between mean and variances in NGD**
>
> We found that this coupled update was a disadvantage, at least numerically. Indeed, the update of the means (the gradient size) is multiplied by the current variance (see Equation 17). Hence, large variances lead to large moves for the mean, and we found this could even lead to mode missing. For instance in the experiment in Figure 3, we needed to initialize our variational posterior with rather large variances (100) to enforce space exploration, and even with reasonable learning rates (0.01) we noticed that behavior. More generally, in our experiments we found that this scheme was more sensitive to the choice of the step size and hence more numerically unstable.
>
> **Q3 : error in the format of references**
>
> Thank you for spotting this - we cleaned it in our revised version.

---

> > ### Comment · Reviewer_Fvva · 2025-08-02
> > **Thanks for the responses**
> >
> > I thank the authors for the comments and maintain my favorable score.

---

### Official Review · Reviewer_pR5Q · 2025-06-23

**Clarity:** 3
**Significance:** 3
**Originality:** 3
**Rating:** 4
**Confidence:** 4

**Summary:**

The paper presents a memory-linear variational-inference scheme that uses an equal-weight mixture of isotropic Gaussians. Two geometry-aware O(d) variance updates—Isotropic Bures-Wasserstein and Entropic Mirror-Descent—combined with gradient steps on the means yield a scalable alternative to full-covariance mixtures, normalising flows, and HMC.

**Questions:**

1. Can you provide guidance or an adaptive strategy for choosing the number of components N?

2. On a synthetic with a 90/10 probability split between two modes, how large is the bias induced by equal weights? Please quantify.

3. More broadly, how does your method mitigate the bias caused by the equal-weight constraint?

4. For the two update methods such as IBW and MD updates, how practitioners choose between them in practice?

**Ethical Concerns:**

["NO or VERY MINOR ethics concerns only"]

**Final Justification:**

This is a decent and well-written paper.
I have no concern of the paper.

**Limitations:**

Authors mention the equal-weight constraint and the memory-accuracy trade-off as limitations. They could further discuss how memory scales if N must grow with dimension and potential biases on highly skewed posteriors.

**Quality:**

3

**Strengths And Weaknesses:**

Quality: The submission is technically sound and claims are well supported by theoretical analysis and experimental results.

Clarity: The submission is clearly written.

Significance: This technique has the potential to improve the SOTA of variational inference.

Orginalty: The new formuation of using an equal-weight mixture of isotropic Gaussians is a balance between computation efficiency and representation power.

Strenghs:

1.	The algorithm has a transparent O(d) complexity per component.

2.	The writing is clear and the theoretical derivations are both rigorous and easy to follow.

3.	Experiments are comprehensive, extending up to a large-scale BNN.

Weakness:

1.	No guidance is given for choosing the number of components N. A sensitivity study is also missing.

2.	The equal-weight assumption may introduce significant bias for skewed posteriors, yet no benchmark quantifies this effect.

---

> ### Author Rebuttal · Authors · 2025-07-31
>
> We thank the reviewer for his careful review and positive comments on our paper. We answer his questions below.
>
> **Q1: guidance or adaptive strategy for choosing N (number of components in the variational MOG approximation)**
>
> In general, we found that the larger the N, the better the result. For instance in Figure 1, while the target has 5 components, a larger N (e.g. 10, 20) enables to get a lower error than N=5. Intuitively, even with a large N, optimizing the variances enables to capture quite general shapes (see also Appendix E for more experiments, e.g. E.3 for 2D visual examples).
> This observation is in accordance with our theoretical results (see end of Section 5 and Appendix D) that states that as N goes to infinity, the approximation error within a MOG family with N components goes to zero (assuming the algorithm achieves this error, i.e. the latter finds the minimizer within the family of N-MOG, this would explain why “the larger N the better”). This reflects the fact that for a given number of components, this family of MOG is rich enough (in particular shaping the variances is crucial) to fit a wide variety of targets.
> In practical applications, this trade-off must be carefully balanced by the user.
> Moreover, the number of modes in the target distribution is typically unknown in advance. Yet, for BNN for instance, we noticed that even with a low number of modes (N=5), the variational approximation of such complex posteriors already leads to very satisfying accuracy results.  Yet (see answer to Q2 of Reviewer cDhA for more results on BNN and the effect of choosing different number of modes N), for these typically multimodal posterior, we saw that increasing N enables to better cover the target density and we are less likely to miss a mode.
>
> **Q 2-3 : about the bias when using constant weights in MOG, and its mitigation**
>
> We conducted experiments on a target mixture of Gaussians in two dimensions with two components and weights [0.9, 0.1] as requested by the reviewer. The means were sampled from a ball of radius 5, and covariances were scaled by a factor of 2. We benchmarked against the algorithms presented in the paper and the baselines used throughout (BW, IBW, MD, NGD) for N = 2, 3, 5, 10 components. As expected with N = 2, since the weights are not optimizable, we observe significant bias induced by the uniform weights constraint. The approximation either misses the low-density mode or assigns equal probability to both modes. With N = 3 already, the bias begins to diminish as the method allocates 2 Gaussians near the heavy mode and 1 near the light mode (with N=5, respectively 4 and 1). When N = 10, we achieve probabilities close to the target distribution, with approximately 9 Gaussians clustering around the heavy mode and 1 around the light mode.
> Methodology: We drew B = 10,000 samples from both the variational distribution and the target distribution, then applied K-means clustering with n_clusters = 2. We analyzed the cluster assignments and proportions. The reported errors are: (1) the average Euclidean distance between the target modes and the variational modes, and (2) the average absolute difference between the target cluster weights and variational cluster weights. The first error captures whether the variational approximation has found the right modes, while the second one measures whether it has found them in the right proportions.
> For readability in our table we round coordinates to one decimal place and probabilities to two decimal places. Our numerical results are the following.
>
> Target: 0.10 at (-4.6, 0.6); 0.90 at (4.7, -2.2)
> - IBW
>     - N = 2: 0.50 at (3.6, -2.2); 0.50 at (5.8, -2.2), Errors: 4.8, 0.4
>     - N = 3: 0.33 at (-4.6, 0.5); 0.67 at (4.7, -2.2), errors: 0.02, 0.23
>     - N = 5: 0.20 at (-4.6,0.5); 0.80 at (4.7, -2.2), errors: 0.05, 0.10
>     - N = 10: 0.10 at (-4.6, 0.6); 0.90 at (4.7, -2.2), errors: 0.02, 0.00
> - MD
>     - N = 2: 0.50 at (3.6, -2.2); 0.50 at (5.8, -2.2), errors: 4.9, 0.4
>     - N = 3: 0.33 at (-4.6, 0.5); 0.67 at (4.7, -2.2), errors: 0.33, 0.30
>     - N = 5: 0.20 at (-4.6,0.5); 0.81 at (4.7, -2.2), errors: 0.05, 0.10
>     - N = 10: 0.10 at (-4.6, 0.6); 0.90 at (4.7, -2.2), errors: 0.05, 0.01
> The detailed table will be included in the revised paper along with additional benchmarks and visualizations. Note that a similar experiment was already conducted in Appendix E, Figure 6, c-3 with 4 modes and different weights, but without quantifying the bias induced by the equal weights restriction.
> To conclude, indeed for a fixed N there might a bias induced by the fixed weights, but the latter bias is easily mitigated by increasing N.
>
> **Q4: choice of IBW vs MD**
>
> See answer to Q4 of Reviewer cDha.

---

> > ### Comment · Reviewer_pR5Q · 2025-08-01
> >
> > I have read the response. Thanks for the clarification.
> > I will keep my score.

---

### Official Review · Reviewer_cDhA · 2025-06-28

**Clarity:** 3
**Significance:** 3
**Originality:** 3
**Rating:** 4
**Confidence:** 4

**Summary:**

The authors present an algorithm for optimizing $KL(q\mid\pi)$ where $q$ denotes a mixture of $N$ isotropic Gaussians, and $\pi$ denotes a target distribution. The algorithm uses a gradient descent step for updating the means and either a Bures or entropic mirror descent update for the variances. The proposed approach, in both flavors, is compared to existing methods for VI with mixtures of Gaussians on UCI regression tasks (Bayesian logistic regression) and MNIST (Bayesian neural network). Additionally, 2d and 20d mixtures of Gaussians are used as a target for comparison of the methods.

**Questions:**

Below are my suggestions that I think would improve the paper:

1. Where are the HMC results in figure 1? They appear in the legend, but I cannot see the line in the figure. I also do not see details on HMC for this experiment in the text. Also the color scheme seems to have additional errors, where is ADVI and what method does the purple line represent? Can the authors please correct me if I'm missing something or comment on the issues in the figure and what it looks like after being fixed?

2. In the "Bayesian posteriors" experiments, I think comparisons with standard Bayesian neural network tools would be appropriate, for example a Laplace approximation (KFAC [1], diagonal, low-rank, etc.) or SWAG [2]. As it stands, we see some evidence that the proposed algorithm (both flavors) does well compared to the included baselines but naturally the question is, "will the standard unimodal approaches do just as well?". Have the authors done this comparison? I understand the BNNs example may not be the focus, but it has been included so the standard baselines should likely be included too.

3. The authors emphasize the value of having a unique variance for each member of the mixture, as opposed to Huix et al. 2024, but I don't see shared variances as a baseline in the experimental results. Can the authors please correct me if I'm wrong or explain why this baseline is missing and if they plan to include it?

4. Why propose two variants of algorithm 1? I'm sure I'm missing something, but I don't see the benefit of having the algorithm contain two options when they perform similarly on all experiments.

Right now my score is "reject" as I think 1-3 suggest the proposed method hasn't been explored enough yet. The writing is quite strong and I enjoy the ideas so I'm happy to increase my score if the above items are addressed. For 2 I think it would be sufficient to compare with a diagonal and a KFAC Laplace approximation since they are good unimodal and expressive unimodal baselines in my mind.

**References**

[1] Ritter, Hippolyt, Aleksandar Botev, and David Barber. "A scalable laplace approximation for neural networks." 6th international conference on learning representations, ICLR 2018-conference track proceedings. Vol. 6. International Conference on Representation Learning, 2018.

[2] Maddox, Wesley J., et al. "A simple baseline for bayesian uncertainty in deep learning." Advances in neural information processing systems 32 (2019).

**Ethical Concerns:**

["NO or VERY MINOR ethics concerns only"]

**Final Justification:**

I feel I still have remaining concerns which make me less confident in the paper. I will increase to borderline reject to reflect the hard effort by the authors and their willingness to engage in the response period. My remaining issues are:
- my initial issue with figure 1 was that the baselines seemed mislabeled but now it seems they were simply not included in the first place so I am less confident in the results presented being representative of what people will see if they use this approach.
- the authors made a comment in the responses about using larger networks than used in their experiments and upon a second look at the paper after this comment I noticed the experiments were on the small side and this comment was not justified in my mind.

I believe these issues are sufficient reason to reject a paper from neurips but I am not going to protest if this paper is accepted.

**Quality:**

2

**Strengths And Weaknesses:**

Strengths:
- The paper is exceptionally clear and well crafted.
- The background for the separate variants of algorithm 1 provides good context and motivates
- I like the framing of the problem and the proposed solution.

Weaknesses:
- Incomplete numerical results and missing baselines, see 2 and 3 in questions.
- Incomplete / confusing figures, see 1 in questions.
- Unclear design choices, see 4 in questions.

---

> ### Author Rebuttal · Authors · 2025-07-31
>
> We thank the author for his careful review and address his main concerns below.
>
> **Q1:  legend of Fig 1**
>
> The reviewer is right, the legend of Fig 1 contains a typo  - ADVI and HMC were baselines included in a later experiment (Figure 4 and 10 on breast_cancer, wine and boston (RMSE) datasets) - we corrected it in the revised version.
>
> **Q2: additional unimodal (N=1) baselines for BNN**
>
> We acknowledge that in the case of BNN, we did not compare with the monoGaussian case (N=1, where N is the number of components in the mixture of Gaussians -MOG- variational posterior). We originally thought that our previous experiments for simple MOG in 2d were already highlighting the critical failure of using one mode for multimodal targets. Note that BNN posteriors are likely to be multimodal (see Figure 3 in [1] for instance). Yet, we acknowledge that including the monoGaussian case as a naive baseline for BNN would have been interesting. We have run these experiments and provide our results below.
>
> We observe that a MOG with N>1 performs better in terms of accuracy and log-likelihood than a mono-Gaussian. Note that we only report the results on one dataset of our algorithms and the requested baselines for readability; we will add complete convergence curves and benchmarks in the revised paper.
>
> On MNIST: Accuracy (higher the better) and negative log-likelihood (lower the better) after convergence on the test set:
> -  IBW
>     - N=1: Acc. 0.872, Neg. LL. 1.594
>     - N=5: Acc. 0.962, Neg. LL. 1.508
> - MD
>     - N=1: Acc. 0.871, Neg. LL. 1.596
>     - N=5: Acc. 0.961, Neg. LL 1.509
> - Laplace Approximation
>     - Diag: Acc. 0.929, Neg. LL . 1.779
>     - KFAC: Acc. 0.948, Neg. LL 1.683
>
> These results show that there is a clear improvement from going to the unimodal (N=1) to multimodal (N>1) VI posterior. We believe this may be due to the fact that multimodal variational posterior enable to explore other basins/modes of good weights for the neural network (see upcoming paragraph) which enhance good generalization, as illustrated in Fig 3 in [2].
> We thank the reviewer for the suggestion and will include these experiments in the paper.
>
> We have also investigated numerically whether our algorithms find different modes.  For this analysis, we used two metrics to assess the diversity of the optimized mixture components. We applied PCA to all collected means from different methods (IBW, MD, NGD, and MAP) and visualized them in a scatter plot (which will be included in the revised paper). Here, MAP refers to the Maximum A Posteriori estimate, which corresponds to the mean of the variational Gaussian approximation used with the Laplace method. Note that the MAP estimate is identical for both diagonal and KFAC variants of the Laplace Approximation, as these methods differ only in their covariance matrix approximations. Since we are not able to include the plot here, we provide the average Euclidean distance between distinct mixture component means in the PCA-projected space. We also provide average cosine similarity between component vectors in the original high-dimensional parameter space (excluding self-similarity). We find:
> - IBW with N = 5: pairwise Euclidean average 455, cosine-similarity: 0.034
> - MD with N = 5: pairwise Euclidean average 455, cosine-similarity: 0.034
>
> The identical metrics between IBW and MD reflect the fact that these methods converge to very similar solutions (see answer to Q4). The low cosine similarity (0.034) indicates that components are nearly orthogonal in the high-dimensional parameter space, confirming the effectiveness of our multimodal approach for capturing complex posterior distributions in BNNs. We will include this experiment.
> Overall, these results demonstrate that mixture components maintain substantial diversity and do not collapse to identical solutions, and enable to get better results than a unimodal variational posterior.
>
> **Q3: about shared-variance (within MOG components) baseline**
>
> Huix et al. 2024 proposed an algorithm that updates a uniform sum of Diracs convolved with a Gaussian kernel of variance epsilon . In their algorithm, the **epsilon parameter is fixed**, and they update only the Dirac positions. Typically, we found numerically at the very beginning of our study that only optimizing the means could lead to situations where the algorithm was trapped in local minimas, and mean particles would stay stuck far from the target. This was a big motivation to optimize the variances as we saw it fixed that problem. Yet, indeed you are right that it is possible to consider as a baseline a version of our algorithms, IBW and MD, where this epsilon parameter would be identical between components (while our schemes, in their full generality, allow different isotropic variances). There are several reasons why we did not present the case of optimizing this shared epsilon:
> - Low computational cost: since epsilon is a 1D parameter, updating it scales with N rather than with d, so updating it is not computationally costly. As an example, for BNNs with millions of parameters, it represents only 5 parameters when N = 5, which is negligible.
> - Readability reason: at the beginning of our numerical study, we conducted experiments with this approach (shared epsilon within the MOG) and found it performed better than not updating epsilon (i.e. the scheme of Huix et al 2024), but worse than having a different epsilon for each Gaussian component in the mixture. Since we needed to make choices about benchmarking—we are comparing with Huix et al.'s method, the full covariance approach of Lambert et al., the NGD method from Lin et al., and two versions of our own method (plus a Normalizing Flow as a general non-specific benchmark)—we chose the clearer, more readable option. Yet, we acknowledge that it could be another useful baseline. We have run such baseline and obtained more results that we present below. We report the percentage of improvement achieved by using different epsilon values for each component compared to sharing a single epsilon across the variational mixture. The percentages below are averaged over 5 experiments.
>     - Figure 1 Toy 2D MOG (IBW / MD):
>         - N=5: 28.9% / 42.3%
>         - N=10: 64.3% / 83.3%
>         - N=20: 66.7% / 100.0%
>     - Figure 6 (a) Toy 2D Funnel (IBW / MD):
>         - N=5: 34.6% / 35.2%
>         - N=20: 48.0% / 62.3%
>         - N=40: 50.0% / 56.0%
>     - Toy Funnel 10D (IBW / MD)
>         - N=5: 27.8% / 18.7%
>         - N=20: 28.5% / 21.9%
>         - N=40: 31.6% / 28.2%
> We will add examples of optimized Gaussians in 2D in the revised paper to demonstrate the impact and importance of having a different variance for each Gaussian component. Indeed for 2D toy examples we are able to visualise optimized Gaussians (represented by a circle) of very different size, and we see it is really beneficial for distribution as the Funnel’s one or skewed one (a, b-1, b-2 in Fig 6).
>
> **Q4: about why we proposed and included two different algorithms, IBW vs MD**
>
> We acknowledge that the two algorithms obtain similar final performances, and that our study does not designate (yet?) a clear winner. We included both schemes because both are novel - (while one is an extended version of Lambert et al 2022) but they significantly differ from the competitors (eg NGD). Note also that the two schemes have the same computational cost O(N(d+1)). Both algorithms minimize the same objective but with respect to different geometries. In standard optimization, it is well known that choosing an appropriate geometry can enhance the speed of convergence (for instance, if the objective is smooth and convex in the chosen geometry, see [3]). This is also known on the space of measures ([4], [5]). In our case, it is not clear in advance whether the VI objective is more suited to the Wasserstein or entropic mirror descent geometries. Such investigation is among what we mentioned in the future work in the conclusion, but we believe this requires  a significant theoretical study. Note also that this VI objective (and its smoothness, convexity) is dependent both on the VI family (mog with isotropic Gaussians) but also the log-posterior target (which in practice in bayesian inference applications, depends on the loss, the data, and the model). Recall that for instance, KL(.|pi) is convex in the Wasserstein-2 geometry when pi is log concave on the family of mono Gaussians. Hence, our goal was to investigate empirically the performance of these two schemes for a wide range of posteriors and hyperparameters within the VI family (e.g. number of components). Note that while the algorithms obtain the same final performance (i.e. they find approximately the same solution), their trajectories are different (because they use  different geometries). This, we have noticed in our experiments early in our numerical study, by tracking the evolutions of the variances for each scheme, and these trajectories were generally different. We will include such plots illustrating the (different) behavior of our schemes in the revised version for more clarity.
>
>  We hope our responses have addressed the reviewer’s concerns and that they will consider revising their score.
>
> [1] Izmailov, P., Vikram, S., Hoffman, M. D., & Wilson, A. G. G. (2021). What are Bayesian neural network posteriors really like?. ICML.
>
> [2] Wilson, A. G., & Izmailov, P. (2020). Bayesian deep learning and a probabilistic perspective of generalization. Neurips.
>
> [3] Lu, H., Freund, R. M., & Nesterov, Y. (2018). Relatively smooth convex optimization by first-order methods, and applications. SIAM Journal on Optimization.
>
> [4] Aubin-Frankowski, P. C., Korba, A., & Léger, F. (2022). Mirror descent with relative smoothness in measure spaces, with application to sinkhorn and EM. Neurips.
>
> [5] C Bonet, T Uscidda, A David, PC Aubin-Frankowski, A Korba. (2024) Mirror and preconditioned gradient descent in wasserstein space. Neurips

---

> > ### Comment · Reviewer_cDhA · 2025-08-04
> > **Lack of complete comparisons can't be overlooked**
> >
> > I appreciate the authors taking the time to give such a detailed response. However, I am still not satisfied with the numerical comparisons being made and will not raise my score. My point 1 was not addressed enough for me to feel comfortable with the numerical comparisons being useful for assessing the value of this approach. The BNNs examples are also lacking in that they use 1 layer neural networks, but claims about nets with millions of parameters are now being made. I understand compute is always a challenge but including BNNs as a topic necessitates more relevant networks be explored.

---

> > > ### Author Response · Authors · 2025-08-04
> > > **Clarification on Addressed Reviewer Points and Final Score**
> > >
> > > Dear Reviewer,
> > >
> > > Thank you for your follow-up. However, we were quite surprised by your final response, as it appears inconsistent with the evaluation criteria you originally stated.
> > >
> > > In your initial review, you wrote:
> > >
> > > > *"Right now my score is ‘reject’ as I think 1–3 suggest the proposed method hasn't been explored enough yet. The writing is quite strong and I enjoy the ideas so I'm happy to increase my score if the above items are addressed. For 2 I think it would be sufficient to compare with a diagonal and a KFAC Laplace approximation since they are good unimodal and expressive unimodal baselines in my mind."*
> > >
> > > In our rebuttal, we specifically focused on addressing these three points:
> > >
> > > - **For point 1**, we acknowledged and corrected the typo in the legend of Figure 1, and clarified where HMC and ADVI actually appear (Figures 4 and 10).
> > >
> > > - **For point 2**, we added exactly the baselines you requested: diagonal and KFAC Laplace approximations, as well as a mono-Gaussian baseline (N = 1) in the BNN setting. We showed clear and consistent performance gains from moving to a multimodal variational posterior.
> > >
> > > - **For point 3**, we implemented a shared-variance version of our algorithm, similar in spirit to Huix et al. 2024, and reported significant drops in performance compared to our full method. We quantified these effects across a range of settings.
> > >
> > > Given this, we were surprised to see your final response continue to reject the paper based on concerns that either (a) were already addressed, or (b) were **not raised in the original review**. For instance:
> > >
> > > > *“The BNNs examples are also lacking in that they use 1 layer neural networks, but claims about nets with millions of parameters are now being made.”*
> > >
> > > This is the first mention of network depth as a concern. Had it been raised earlier, we would have considered additional experiments. More importantly, your remark about “claims” of networks with millions of parameters appears to stem from a misunderstanding of a sentence in our rebuttal:
> > >
> > > > *“As an example, for BNNs with millions of parameters, it represents only 5 parameters when N = 5, which is negligible.”*
> > >
> > > We agree that this sentence was **poorly phrased** and potentially misleading. What we intended to convey was this:
> > >
> > > - In our BNN experiments, the two-layer MLP already has over **200,000 parameters** (28×28 input, 256 hidden units, 10 output classes).
> > > - Since we use a mixture of **N = 5** isotropic Gaussians as a variational posterior, we are optimizing **5 mean vectors** of dimension >200k and **5 scalar variances**, yielding over **1 million variational parameters** in total.
> > > - The point was purely about **computational scaling in the variational family**, not about the size of the BNNs used in our experiments.
> > > We regret the phrasing and will correct it in the revised version of the paper.
> > >
> > > To conclude: we fully respect your right to maintain your score, but we feel compelled to point out that (1) we addressed your originally stated concerns comprehensively and with new experiments, and (2) the final justification for rejection appears to shift the evaluation criteria post hoc. We would appreciate it if the review could reflect the work we put into engaging constructively with your feedback.

---

> > > > ### Comment · Reviewer_cDhA · 2025-08-06
> > > > **Acknowleding effort and clarifying opinion**
> > > >
> > > > I would like to start by saying that I absolutely acknowledge and appreciate the time the authors put into responses. I know this is time consuming and the work put in is reflected by a well crafted paper that reads cleanly. I had no intention to downplay your effort but effort alone is not what's at play here. Below I write out my thinking more directly.
> > > >
> > > > The comment I made was primarily due to the wording used and the experiments completed. It doesn't seem appropriate to discuss networks with millions of parameters if none were used in these experiments. Revisiting the numerical results after this claim made me realize how small scale the exploration was wrt network size so I became less confident in the results. For my q1, the conclusion seemed to be there were less baselines used than I initially saw in the figure–this made me less confident in the results.

---

> > > > > ### Author Response · Authors · 2025-08-07
> > > > > **Final clarification**
> > > > >
> > > > > Dear Reviewer,
> > > > >
> > > > > Thank you for your latest comment.
> > > > >
> > > > > We’d like to make one final clarification for the record. The remark about “BNNs with millions of parameters” in our **rebuttal** referred specifically to the dimensionality of the variational parameters, not the size of the network architecture. Our two-layer MLP used in the BNN experiments already has over 200,000 parameters (28×28 input, 256 hidden units, 10 outputs). With N = 5 mixture components, we optimize 5 mean vectors in this high-dimensional space, totaling over 1 million variational parameters. We acknowledge that our sentence in the rebuttal was poorly phrased and regret any confusion caused.
> > > > >
> > > > > We also recall that our experimental setup is already larger in scope than related work/our competitors using mixtures of Gaussians for variational inference. Indeed:
> > > > >
> > > > > - Lin et al. (2019) used 1-layer neural networks with 50 hidden units on small UCI datasets;
> > > > >
> > > > > - Lambert et al. (2022) only evaluated on Bayesian logistic regression;
> > > > >
> > > > > In contrast, we applied all methods to BNNs on MNIST, and incorporated a thorough comparison with multiple baselines.
> > > > >
> > > > > More importantly, we respectfully note that your final decision appears to rely on new concerns — namely, that the size of the BNN in our experiments is "too small" — which were not mentioned in your original review. That review gave clear and specific feedback: (1) clarify the figure, (2) compare with diagonal and KFAC Laplace baselines, and (3) include a shared-variance baseline. We addressed each point directly, with new experiments and quantitative results.
> > > > >
> > > > > We understand if you ultimately remain unconvinced by the contribution. However, we feel it's important to document that the criteria for evaluation changed after rebuttal, and that the final assessment was based on points not originally raised.

---

### Official Review · Reviewer_M74f · 2025-06-30

**Clarity:** 3
**Significance:** 2
**Originality:** 2
**Rating:** 4
**Confidence:** 3

**Summary:**

The paper studies variational inference when the variational family consists of mixtures of isotropic Gaussians where each component has the same weight. The authors propose optimization algorithms and perform numerical experiments that show comparable performance to existing methods with lower memory cost.

**Questions:**

For the results shown in Figure 2, Was the target a Gaussian mixture with 10 components and d=20? Why not show the comparison for multiple values of N, not just 15?

**Ethical Concerns:**

["NO or VERY MINOR ethics concerns only"]

**Final Justification:**

I have read the authors' rebuttal and will keep my score.

**Limitations:**

The paper would have been stronger with some of the items listed as future works.

**Quality:**

3

**Strengths And Weaknesses:**

**Strengths**

- The topic is interesting, mixtures of isotropic Gaussians are complex enough to solve interesting problems, but simple enough to be relatively efficient.

- The definitions and assumptions are stated clearly.

- The authors propose two optimization algorithms and compare their results to exiting works.


**Weaknesses**

A weakness of this paper is the lack of theoretical guarantees on the approximation error of their scheme vs a full covariance matrix, which the authors mention as future work.

The numerical experiments show performance similar to BW, which optimizes full covariance matrices. Except for the gain in memory, I believe the only reported gain in computational cost is the one shown in Figure 2.

---

> ### Author Rebuttal · Authors · 2025-07-31
>
> We thank the reviewer for his positive comments on our paper. We address his main concerns below.
>
> **Weakness 1: lack of theoretical guarantees on the approximation error of our schemes vs full covariance matrix**
>
> With all due respect, we believe there may have been a misunderstanding in the reviewer’s comment regarding “the theoretical guarantees on the approximation error of their scheme vs a full covariance matrix.”
> Specifically, the term *approximation error* refers to the minimal KL divergence achievable within a given variational family, which is independent of the optimization scheme. In contrast, *optimization error*—or equivalently, convergence guarantees—is indeed dependent on the chosen optimization method. We will clarify this distinction more explicitly in the revised manuscript.
>
> - Regarding establishing **approximation guarantees** for VI within a family of mixtures of Gaussians, this is a fundamentally hard problem. Recent guarantees have been obtained in Huix et al 2024 for MOG with constant weights and identical (shared) isotropic covariance matrix, and we have been extending this result (inheriting the same upper bound in logN/N, where N is the number of components) with different covariance matrices (either **isotropic or full**), see end of Section 5 l262 which refers to our result in Appendix D. Yet, this common bound (valid for both cases - isotropic or full) does not enable to capture the small increase in KL objective we notice in our experiments when comparing isotropic to full covariance matrices; which is why we mention in future work that studying further the approximation error is worth pursuing, in particular establishing whether this rate is tight (by proving a lower bound) is an open research question to the best of our knowledge.
>
> - Regarding establishing **convergence guarantees** for our IBW (isotropic Bures-Wasserstein) and MD scheme, we acknowledge that we lack such guarantees in the general case (N>1 and general target distribution). We would like to emphasize that the original paper Lambert et al 2022, which introduced the method employing full covariance matrices for mixtures of Gaussians (namely BW in our paper), which is the full covariance matrix extension of our scheme IBW, also did not provide theoretical guarantees for N>1. This reflects a broader challenge in the field of VI with mixtures of Gaussians: outside the convex setting—such as a single Gaussian approximating a log-concave target—obtaining rigorous approximation error bounds becomes significantly more difficult since we move into the non-convex regime. This is a well-known limitation across related methods, not unique to our approach,  and remains an open problem in general.
>
> We provided global convergence guarantees (in continuous time) only for N=1 and when the posterior is strongly log-concave (Prop 2.1). Indeed, the KL is convex in the Wasserstein geometry, and isotropic Gaussians are preserved along Wasserstein geodesics. However, when the posterior is not log-concave and N>1, this structure is lost, and we face a non-convex problem. In such settings, the most one can hope for are descent-type results: that the KL objective decreases along IBW iterations; see for instance Prop 4, Huix et al. 2024 for mixtures of isotropic Gaussians with shared covariance, where the VI objective is shown to be smooth w.r.t. the Wasserstein geometry. For our MD scheme, we would need to establish similar descent results, but this time relying on relative smoothness of the objective with respect to another geometry induced by the KL as a Bregman divergence, see for instance (see [1]). However, establishing such results for both our schemes require a strong technical study that we believe can be deferred to future work.
>
> [1] Aubin-Frankowski, P. C., Korba, A., & Léger, F. (2022). Mirror descent with relative smoothness in measure spaces, with application to sinkhorn and em. Neurips.
>
> **Weakness 2: lack of illustration of the gain in memory for our isotropic variance schemes vs full covariance case**
>
> We acknowledge that Figure 2 is the sole empirical demonstration of our computational advantage; however, we believe it effectively captures the core trade-off. It illustrates the scaling behavior of both the KL objective and running time across varying dimensions, revealing a clear transition in computational cost **from linear (IBW) to quadratic (BW)** with respect to dimensionality. This scaling behavior is theoretically predictable, as emphasized throughout the paper.
> The observed difference in computational complexity directly stems from the reduction in parameter space—from O(N(d + d²)) to O(N(d + 1))—when using the isotropic family. As dimensionality increases, the computational benefits of this reduction become more pronounced, while the gap in KL divergence remains small. This is precisely what makes our method attractive for high-dimensional settings.
> In such regimes, full covariance methods often become computationally infeasible on standard hardware due to their memory and runtime demands—a limitation that is theoretically inevitable given their quadratic scaling. Rather than serving as standalone evidence, Figure 2 validates our theoretical predictions. While we are happy to provide further experiments, we believe this figure already conveys the central insight: our method offers comparable approximation quality at a fraction of the computational cost, enabling scalable inference in high dimensions.
>
> **Question about Figure 2**
>
> We realised that our formatting was confusing. We apologize for this and will update the organization of the figures for greater clarity. The issue stems from our layout: we discuss Figure 3 first, followed by Figure 2, but the description of Figure 3 appears to the left of Figure 2. This spatial arrangement made it unclear which figure we were describing, and we believe the reviewer was referring to Figure 3, not Figure 2.
>
> In Figure 3, the goal of the experiment is to demonstrate that our methods (IBW, MD) approximates the target mixture as well as  the full covariance version (BW) of the algorithm. We also wanted to highlight that NGD suffers from mode collapse.
> While we could have conducted this analysis for different N values, that was not the goal of these particular experiments, which was to show that our schemes were working in high dimensions for a relatively large N (20). Also, we illustrated the effect of N in Figure 1 (and 6 in the Appendix), so for readability reasons we chose not to show the comparison with multiple N. But we agree with the reviewer that investigating the role of this hyperparameter is interesting, and we will add more plots on the same experiments with different N values in the appendix.

---

> > ### Comment · Reviewer_M74f · 2025-08-01
> >
> > I thank the authors for the detailed response and clarifications. I am satisfied with the authors' answers and I will keep my score.

---

### Decision · Program_Chairs · 2025-09-17

**Decision:**

Accept (poster)

**Comment:**

This paper addresses a restricted variational family, i.e., mixture of isotropic Gaussians with uniformly fixed weights and demonstrates its validity and usefulness in the context of expressivity vs memory-efficiency. Two geometry-aware variance update rules, which are decoupled from the mean update, are presented in the context of mixtures of isotropic Gaussians. Most of reviewers feel that the paper is well written and has interesting contributions. However, there are some concerns in approximation/convergence guarantees and empirical comparisons. During the rebuttal period, some concerns were clarified but the concern on experiments with large scale models was not fully resolved. Nevertheless, thanks to extensive communications with reviewers, we have reached consensus on this paper.